# FAST, EXPRESSIVE $\mathrm{SE}(n)$ EQUIVARIANT NETWORKS THROUGH WEIGHT-SHARING IN POSITION-ORIENTATION SPACE

**Erik J. Bekkers**[1*], **Sharvaree Vadgama**[1*]
**Rob D. Hesselink**[1], **Putri A. van der Linden**[1], **David W. Romero**[2,3]
[1] University of Amsterdam    [2] Vrije Universiteit Amsterdam    [3] NVIDIA Research

[*]*Equal contribution.*      *Code available at https://github.com/ebekkers/ponita*

## ABSTRACT

Based on the theory of homogeneous spaces we derive *geometrically optimal edge attributes* to be used within the flexible message-passing framework. We formalize the notion of weight sharing in convolutional networks as the sharing of message functions over point-pairs that should be treated equally. We define equivalence classes of point-pairs that are identical up to a transformation in the group and derive attributes that uniquely identify these classes. Weight sharing is then obtained by conditioning message functions on these attributes. As an application of the theory, we develop an efficient equivariant group convolutional network for processing 3D point clouds. The theory of homogeneous spaces tells us how to do group convolutions with feature maps over the homogeneous space of positions $\mathbb{R}^3$, position and orientations $\mathbb{R}^3 \times S^2$, and the group $\mathrm{SE}(3)$ itself. Among these, $\mathbb{R}^3 \times S^2$ is an optimal choice due to the ability to represent directional information, which $\mathbb{R}^3$ methods cannot, and it significantly enhances computational efficiency compared to indexing features on the full $\mathrm{SE}(3)$ group. We support this claim with state-of-the-art results –in accuracy and speed– on five different benchmarks in 2D and 3D, including interatomic potential energy prediction, trajectory forecasting in N-body systems, and generating molecules via equivariant diffusion models.

## 1 INTRODUCTION

Inspired by the foundational role of convolution operators in deep learning and the 'convolution is all you need' theorem (Cohen et al. (2019, Thm 3.1); Bekkers (2019, Thm 1)) -which asserts that any layer that is linear and equivariant must be a group convolution, we propose an efficient and expressive group convolutional approach for constructing neural networks equivariant to $\mathrm{SE}(n)$: the group of $n$-dimensional translations and rotations. While this theorem is a theoretical result, several studies provide empirical truth to the statement as well. For example, ConvNeXt (Liu et al., 2022b) challenges the need for Transformers (Vaswani et al., 2017) in vision tasks, and Romero et al. (2021); Poli et al. (2023) show that convolutions are sufficient to model long context in sequences, e.g., language, without the need for transformers or recurrent networks.

In our work, we revisit the influential inductive bias of *weight sharing* in convolutions (LeCun et al., 1998), classically defined as the sharing of a convolution kernel (linear transformation) over all the neighborhoods in an image. In the discrete image setting, the kernel is given as a set of weights and it is appropriate to refer to the convolution kernel as *the weights*. However, in the continuous case, and more general processing frameworks such as in message passing networks (MPNs) (Gilmer et al., 2017a), this terminology no longer literally applies. Therefore, we formalize the notion of weight sharing to gain insights into how to use this inductive bias in a more general setting.

Traditionally, weight sharing refers to the idea of using the same linear transformation matrix over neighborhoods identical up to translation. We generalize this notion by the construction of equivalence classes of neighboring point-pairs, in which we say neighbor pairs are equivalent if they are identical up to a transformation in a group $G$. Weight sharing then becomes the notion of sharing message functions in MPNs over the equivalence classes, which is achieved by conditioning the functions on *attributes* that act as identifiers for the equivalence classes. In Sec. 3.1 we formalize this construction and present all the attributes one needs for $\mathrm{SE}(n)$ equivariant weight sharing.

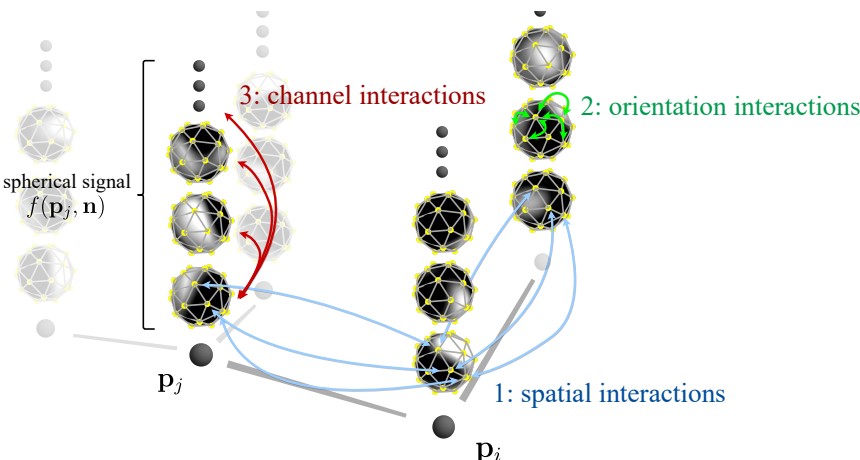

Figure 1: Separable group convolutions on position orientation space $\mathbb{R}^3 \times S^2$. Efficiency is obtained due to parallelizing the most expensive step, step 1 (message passing), over orientations and channels. Steps 2 and 3 are efficient as well as spherical convolutions are batched over positions and channels, and channel mixing is batched over positions and orientations.

As an application of the theory, we construct expressive and efficient $\mathrm{SE}(n)$ equivariant group convolutional networks for the processing of 3D point clouds. Our simple, fully convolutional architecture achieves state-of-the-art results on three different benchmarks: interatomic potential energy prediction, trajectory forecasting in N-body systems, and generating molecules via equivariant diffusion models. The contributions of this paper are summarized as follows:

- We formalize the notion of weight sharing in Sec. 3.1.
- We derive optimal attributes (Thm. 1) for weight sharing over equivalence classes of point pairs and show that these *are all you need* to build equivariant universal approximators (Cor. 1.1).
- We present a fast, expressive equivariant architecture based purely on convolutions (Fig. 1). Our method reaches state-of-the-art performance on three equivariant benchmarks.

## 2 BACKGROUND

### 2.1 MATHEMATICAL PREREQUISITES

**Groups and homogeneous spaces.** A *group* is an algebraic construction defined by a set $G$ and a binary operator $\cdot : G \times G \to G$, known as the *group product*. This structure must satisfy the following axioms: *closure*, where $\forall_{h,g \in G} : h\,g \in G$; the existence of both an *identity* $e$ and an *inverse* $g^{-1}$ element such that $g^{-1} \cdot g = e$; and *associativity*, where $\forall_{g,h,i \in G} : (g \cdot h) \cdot i = g \cdot (h \cdot i)$. We denote the group product between two elements $g, g' \in G$ by juxtaposition, i.e., as $g\,g'$.

It is useful to think of the elements of $G$ as transformations. The group product then tells how a transformation $g'$ followed by another transformation $g$ can be represented by a single transformation $g\,g'$. We focus on the *Special Euclidean motion group* $\mathrm{SE}(n)$ consisting of distance and orientation-preserving transformations. Elements $g = (\mathbf{x}, \mathbf{R}) \in \mathrm{SE}(n)$ are parameterized by translation vectors $\mathbf{x} \in \mathbb{R}^n$ and rotation matrices $\mathbf{R} \in \mathrm{SO}(n)$. Here, $\mathrm{SO}(n)$ denotes the set of $n \times n$ matrices with determinant 1, which forms a group in itself with matrix multiplication as a group product. The $\mathrm{SE}(n)$ group product between two roto-translations $g = (\mathbf{x}, \mathbf{R})$ and $g' = (\mathbf{x}', \mathbf{R}')$ is given by $(\mathbf{x}, \mathbf{R})(\mathbf{x}', \mathbf{R}') = (\mathbf{R}\mathbf{x}' + \mathbf{x}, \mathbf{R}\mathbf{R}')$, and its identity element is given by $e = (\mathbf{0}, \mathbf{I})$.

A group can act on spaces other than itself via a *group action* $\mathcal{T} : G \times X \to X$, where $X$ is the space on which $G$ acts. For simplicity, we denote the action of $g \in G$ on $x \in X$ as $g\,x$. Such a transformation is called a group action if it is homomorphic to $G$ and its group product. That is, it follows the group structure: $(g\,g')\,x = g\,(g'\,x)\ \forall g, g' \in G, x \in X$, and $e\,x = x$. For example, consider the space of 3D positions $X = \mathbb{R}^3$, e.g., atomic coordinates, acted upon by the group $G = \mathrm{SE}(3)$. A position $\mathbf{p} \in \mathbb{R}^3$ is roto-translated by the action of an element $(\mathbf{x}, \mathbf{R}) \in \mathrm{SE}(3)$ as $(\mathbf{x}, \mathbf{R})\,\mathbf{p} = \mathbf{R}\,\mathbf{p} + \mathbf{x}$.

A group action is termed *transitive* if every element $x \in X$ can be reached from an arbitrary origin $x_0 \in X$ through the action of some $g \in G$, i.e., $x = g x_0$. A space $X$ equipped with a transitive action of $G$ is called a *homogeneous space* of $G$. Finally, the *orbit* $G\,x := \{g\,x \mid g \in G\}$ of an element

$x$ under the action of a group $G$ represents the set of all possible transformations of $x$ by $G$. For homogeneous spaces, $X = G\,x_0$ for any arbitrary origin $x_0 \in X$.

**Quotient spaces.** The aforementioned space of 3D positions $X = \mathbb{R}^3$ serves as a homogeneous space of $G = \mathrm{SE}(3)$, as every element $\mathbf{p}$ can be reached by a roto-translation from $\mathbf{0}$, i.e., for every $\mathbf{p}$ there exists a $(\mathbf{x}, \mathbf{R})$ such that $\mathbf{p} = (\mathbf{x}, \mathbf{R})\,\mathbf{0} = \mathbf{R}\,\mathbf{0} + \mathbf{x} = \mathbf{x}$. Note that there are several elements in $\mathrm{SE}(3)$ that transport the origin $\mathbf{0}$ to $\mathbf{p}$, as any action with a translation vector $\mathbf{x} = \mathbf{p}$ suffices regardless of the rotation $\mathbf{R}$. This is because any rotation $\mathbf{R}' \in \mathrm{SO}(3)$ leaves the origin unaltered.

We denote the set of all elements in $G$ that leave an origin $x_0 \in X$ unaltered the *stabilizer subgroup* $\mathrm{Stab}_G(x_0)$. In subsequent analyses, we use the symbol $H$ to denote the stabilizer subgroup of a chosen origin $x_0$ in a homogeneous space, i.e., $H = \mathrm{Stab}_G(x_0)$. We further denote the *left coset* of $H$ in $G$ as $g\,H := \{g\,h \mid h \in H\}$. In the example of positions $\mathbf{p} \in X = \mathbb{R}^3$ we concluded that we can associate a point $\mathbf{p}$ with many group elements $g \in \mathrm{SE}(3)$ that satisfy $\mathbf{p} = g\,\mathbf{0}$. In general, letting $g_x$ be any group element s.t. $x = g_x\,x_0$, then any group element in the left set $g_x\,H$ is also identified with the point $\mathbf{p}$. Hence, any $x \in X$ can be identified with a left coset $g_x H$ and vice versa.

Left cosets $g\,H$ then establish an *equivalence relation* $\sim$ among transformations in $G$. We say that two elements $g, g' \in G$ are equivalent, i.e., $g \sim g'$, if and only if $g\,x_0 = g'\,x_0$. That is, if they belong to the same coset $g\,H$. The space of left cosets is commonly referred to as the *quotient space* $G/H$.

We consider *feature maps* $f : X \to \mathbb{R}^C$ as multi-channel signals over homogeneous spaces $X$. Such maps are of interest as they often form the hidden representations in various deep learning tasks. In this work, we treat point clouds as sparse feature maps, e.g., sampled only at atomic positions. In the general continuous setting, we denote the space of feature maps over $X$ with $\mathcal{X}$. Such feature maps undergo group transformations through *regular group representations* $\rho^{\mathcal{X}}(g) : \mathcal{X} \to \mathcal{X}$ parameterized by $g$, and which transform functions $f \in \mathcal{X}$ via $[\rho^{\mathcal{X}}(g)f](x) = f(g^{-1}x)$.

## 2.2 Motivation 1: Group convolution is all you need

In deep learning, we often employ learnable operators $\Phi : \mathcal{X} \to \mathcal{Y}$, such as self-attention or convolution layers, to iteratively transform feature maps. Such an operator is termed *G-equivariant* if it commutes with group representations on the input and output feature maps: $\rho^{\mathcal{Y}}(g) \circ \Phi = \Phi \circ \rho^{\mathcal{X}}(g)$. Group equivariance ensures that operators preserve the geometric structure of the data, meaning that derived features follow the same transformation laws as the input.

An important result in the field of equivariant deep learning and signal processing is that if we want $\Phi$ to be linear and group equivariant, then it *must be a group convolution*:

$$[\Phi f](y) = \int_X k(g_y^{-1}x)f(x)\mathrm{d}x\,. \tag{1}$$

Essentially, the group convolution performs template matching of a kernel $k$ against patterns in $f$ by taking $\mathbb{L}_2$-inner products of the shifted kernel $k(g_y^{-1}\cdot)$ and $f$. Recall that $g_y$ denotes any group element such that $y = g_y\,y_0$, and any other group element in the set $g_y\,H$ is also valid. This implies that Eq. 1 is only valid if the kernel is invariant to left actions of $H$, i.e. if $\forall_{h \in H} : k(h^{-1}x) = k(x)$. These findings are summarized in various seminal equivariant deep learning works (Cohen et al., 2019, Thm. 3.1 *convolution is all you need*), (Bekkers, 2019, Thm 1), (Kondor & Trivedi, 2018).

In light of the "convolution is all you need" claim and the pivotal role of convolutions in areas like computer vision and signal processing, one might question the need for anything more complex than convolutions. We therefore explore the potential of a straightforward, yet theoretically grounded, fully convolutional approach to equivariant deep learning.

## 2.3 Motivation 2: Efficiency and expressivity –The Homogeneous space $\mathbb{R}^3 \times S^2$

It is worth noting that the domain of the output signal $Y$ in Eq. 1 does not have to match the domain of the input signal $X$. The aforementioned theorems show that $\mathrm{SE}(3)$ equivariant convolutions on feature maps with domain $\mathbb{R}^3 \equiv \mathrm{SE}(3)/\mathrm{SO}(3)$ require isotropic, rotation invariant kernels, as seen in SchNet (Schütt et al., 2023). However, maximal expressivity is gained when the kernel has no constraints. This is achieved when the domain $Y$ is the group itself, i.e., $\mathrm{SE}(3)$ as $H = \{e\}$ is then the trivial group. Generating higher-dimensional feature maps, from $X = \mathbb{R}^3$ to $Y = \mathrm{SE}(3)$, is typically referred to as *lifting*. It is done by performing template matching (Eq. 1) over both translations and rotations, as opposed to just translations. Despite enhanced expressivity, however, subsequent layers must compute integrals over the whole $\mathrm{SE}(3)$ space, which can be computationally restrictive.

In this work, we opt for a middle ground and define feature maps on the **homogeneous space of positions and orientations** $X=\mathbb{R}^3\times S^2\equiv\mathrm{SE}(3)/\mathrm{SO}(2)$. We denote elements in $\mathbb{R}^3\times S^2$ as tuples $(\mathbf{p},\mathbf{o})$ of positions $\mathbf{p}\in\mathbb{R}^3$ and orientations $\mathbf{o}\in S^2$. We set the origin as $x_0=(\mathbf{0},\mathbf{e}_z)$, where $\mathbf{e}_z=(0,0,1)^\intercal$. Every orientation can be obtained from a rotation $\mathbf{R}_\mathbf{o}\in\mathrm{SO}(3)$ that maps the unit vector $\mathbf{e}_z$ to $\mathbf{o}$. Hence, $S^2$ is a homogeneous space of $\mathrm{SO}(3)$ with stabilizer subgroup $H=\mathrm{Stab}_{\mathrm{SO}(3)}(\mathbf{e}_z)\equiv\mathrm{SO}(2)$: the group of rotations around the $z$-axis. Therefore, each position-orientation tuple corresponds to an equivalence class of transformations in $\mathrm{SE}(3)$, formalized as $(\mathbf{p},\mathbf{o})\equiv(\mathbf{p},\mathbf{R}_\mathbf{o})\,H$ in $\mathrm{SE}(3)/\mathrm{SO}(2)$.

While methods based on $\mathbb{R}^3$ positions are computationally efficient, intermediate feature maps may fail to capture directional information due to kernel constraints. On the other hand, full $\mathrm{SE}(3)$ group convolutions are computationally demanding but excel at capturing directional features. Compared to the latter, our method offers a significant improvement in computational efficiency *without compromising the capability to represent directional features*. This claim is proven in (Gasteiger et al., 2021, Thm. 3) by relying on the fact that equivariant predictions can be obtained by combining an equivariant basis with coefficients obtained from an invariant network (Villar et al., 2021). In DimeNet and GemNet (Gasteiger et al., 2019; 2021), the basis is determined by the direction of edges in an atomic point cloud, and coefficients are derived through *invariant message passing*. Rather than assigning a sparse basis of reference directions in $S^2$ to each point position, our method employs a *dense basis* and assign a *spherical signal* to each point in $\mathbb{R}^3$, see Fig. 1.

### 2.4 MOTIVATION 3: CONDITIONAL MESSAGE PASSING AS GENERALIZED CONVOLUTION

**Message passing.** Our primary focus lies in the processing of point clouds on homogeneous spaces. Point cloud methods are conventionally discussed within the broader context of message passing (Gilmer et al., 2017a). We intentionally adopt this framework as it serves as a familiar and versatile paradigm suitable for the description of various deep learning layers, including convolutions.

We model point-clouds as graphs $\mathcal{G}=(\mathcal{V},\mathcal{E})$, with nodes $i\in\mathcal{V}$ and edges $(i,j)\in\mathcal{E}$. Each node $i$ has an associated coordinate $x_i\in X$ in a homogeneous space $X$. We consider features over such graphs as sparse discretizations of dense feature maps $f:X\to\mathbb{R}^C$ with node features $(x_i,f_i)\Leftrightarrow f(x_i)$. Considering optional pair-wise edge attributes $a_{ij}$, message passing layers update feature nodes as:

1. *compute the message $m_{ij}$ from node $j$ to $i$:* $\qquad m_{ij}=\phi_m\left(f_i,f_j,a_{ij}\right),\qquad$ (2)

2. *Perform a permutation invariant message aggregation:* $\qquad m_i=\sum_{j\in\mathcal{N}(i)}m_{ij},\qquad$ (3)

3. *Update the node features:* $\qquad f_i^{out}=\phi_f\left(f_i,m_i\right).\qquad$ (4)

Here, $\mathcal{N}(i)=\{j\mid(i,j)\in\mathcal{E}\}$ denotes the set of neighbors of the node $i$.

**Convolutional message passing.** Convolutions can be described in the message passing framework as approximations over a sparse set of points on which continuous signals are sampled:

$$\int_X k(g_x^{-1}x')f(x')\mathrm{d}x'\approx\sum_{j\in\mathcal{N}(i)}k(g_{x_i}^{-1}x_j)f_j.\qquad(5)$$

Here, we can recognize the message passing form, in which $\phi_m(f_j,g_{x_i}^{-1}x_j)=k(g_{x_j}^{-1}x_i)f_j$ is a linear transformation performed by matrix-vector multiplication with the kernel $k(g_{x_j}^{-1}x_i)\in\mathbb{R}^{C_{out}\times C}$ that depends on $g_{x_j}^{-1}x_i$. Following Brandstetter et al. (2021), we interpret this as using a message function that is *conditioned on the attribute $a_{ij}$*, denoted as $\phi_m(f_i,f_j;a_{ij})$ to emphasize this dependency. We could directly consider the use of the invariant attribute $a_{ij}=g_{x_j}^{-1}x_i$ in conditional message passing as an extension of convolution. However, it's crucial to recognize that, despite its invariance under global group actions, the value of this attribute depends on the selected representative $g_{x_i}$. Specifically, any $g_{x_i}h$, where $h$ belongs to the stabilizing group $H$, can result in a different $a_{ij}=h^{-1}g_{x_i}^{-1}x_j$. In essence, any $a_{ij}\in H\,g_{x_i}x_j$ within the orbit of $H$ should be treated equivalently.

## 3 FAST, EXPRESSIVE $\mathrm{SE}(n)$ EQUIVARIANT NEURAL NETWORKS

### 3.1 WEIGHT-SHARING AND OPTIMAL INVARIANT ATTRIBUTES

**Problem statement.** Our objective is to find an invariant attribute $a_{ij}$ that can be associated with two points $(x_i,x_j)$ in a homogeneous space $X$ of a group $G$, such that it satisfies these criteria:

    (i) *Invariance to the global action of $G$.* Any pair in the equivalence class $[x_i, x_j] := \{g\,x_i, g\,x_j \mid g \in G\}$ must be mapped to same attribute $a_{ij}$.

    (ii) *Uniqueness.* Each attribute $a_{ij}$ should be unique for the given equivalence class.

This problem boils down to finding a *bijective* map $[x_i, x_j] \mapsto a_{ij}$. Note that invariance alone is insufficient to build an expressive network. For example, a mapping $[x_i, x_j] \mapsto 0$ is invariant but trivial, and thus not useful. Bijectivity ensures that each attribute is unique and it fully characterizes the space of all possible equivalence classes of point pairs. In essence, a bijective attribute mapping is all you need to enable weight-sharing over equivalent point pairs and obtain full expressiveness. The notion of weight sharing is formalized by the following three definitions:

**Definition 3.1** (*Equivalent point pairs*)**.** Two point pairs $(x_i, x_j), (x_i', x_j') \in X \times X$ in a homogeneous space $X$ of a group $G$ are called *equivalent* iff they can be transformed into each other through left multiplication by an element $g \in G$. We define this equivalence relation as:

$$(x_i, x_j) \sim (x_i', x_j') \iff \exists_{g \in G} : (x_i', x_j') = (g\,x_i, g\,x_j).$$

**Definition 3.2** (*Equivalence class of point pairs*)**.** The equivalence relation $\sim$ of definition 3.1 defines an *equivalence class* of point pairs, denoted as:

$$[x_i, x_j] = \{(x_i', x_j') \in X \times X \mid (x_i', x_j') \sim (x_i, x_j)\}.$$

Here, $[x_i, x_j]$ represents a set of point pairs that should be treated as equivalent, with $(x_i, x_j)$ serving as its *representative*. The *space of equivalence classes* of point pairs is denoted as $X \times X / \sim$.

**Definition 3.3** (*Weight-sharing in message passing*)**.** A message passing layer (Eq. 2-4) is said to *share weights* over equivalent point pairs if it processes equivalent point pairs in the same way. That is, if its message function $\phi_m(\mathbf{f}_i, \mathbf{f}_j; [x_i, x_j])$ is conditioned on the equivalence class $[x_i, x_j]$.

See (Koishekenov & Bekkers, 2023) for options for conditioning message functions. Our goal now is to parameterize the space of equivalence classes $X \times X / \sim$ with concrete attributes that condition message functions. To that end, we reduce the problem of identifying equivalence classes of point pairs to the task of identifying orbits of single points $X$, akin to LieConv (Finzi et al., 2020a). We show that equivalence classes $[x_i, x_j]$ correspond to orbits in $H \backslash X$ using the following lemma:

**Lemma 1** (*Equivalence class correspond to $H$-orbits in $X$*)**.** *For any chosen representatives $g_i \in G$ of $x_i \in X \equiv G/H$ such that $x_i = g_i x_0$, and any $x_j \in X$, the following mapping from the space of equivalence classes of point pairs $X \times X / \sim$ to the space $H \backslash X$ of orbits of $H$ in $X$ is a bijection:*

$$[x_i, x_j] \mapsto H g_i^{-1} x_j, \qquad [x_i, x_j] \in X \times X / \sim, \;\; g_i^{-1} x_j \in H \backslash X. \tag{6}$$

*Proof.* See the appendix, section B.1. $\qquad\square$

The subsequent theorem characterizes bijective mappings from equivalence classes $[x_i, x_j]$ to concrete attributes $a_{ij}$ that we can compute for homogeneous spaces of $\mathrm{SE}(n)$. Since these mappings are bijective, the attributes serve as unique identifiers of the equivalence classes. Therefore, they are sufficient for the construction of expressive yet efficient equivariant message passing networks.

**Theorem 1** (*Bijective attributes for homogeneous spaces of $\mathrm{SE}(n)$*)**.** *Consider three pairs of homogeneous spaces $X \equiv \mathrm{SE}(n)/H$ and stabilizer subgroups $H$: (i) position space: $X = \mathbb{R}^n$, $H = \mathrm{SO}(n)$; (ii) position-orientation space: $X = \mathbb{R}^d \times S^{n-1}$, $H = \mathrm{SO}(n-1)$; and (iii) the whole group: $X = \mathrm{SE}(n)$, $H = \{e\}$. For $n = 2, 3$, the following mappings from equivalence classes $[x_i, x_j] \in X \times X / \sim$, as defined in definitions 3.1 and 3.2, to explicit attributes are bijective:*

$$\mathbb{R}^2 \text{ and } \mathbb{R}^3: \qquad\qquad\qquad [\mathbf{p}_i, \mathbf{p}_j] \;\mapsto\; a_{ij} = \|\mathbf{p}_j - \mathbf{p}_i\|, \tag{7}$$

$$\mathbb{R}^2 \times S^1 \text{ and } \mathrm{SE}(2): \quad [(\mathbf{p}_i, \mathbf{o}_i), (\mathbf{p}_j, \mathbf{o}_j)] \;\mapsto\; a_{ij} = (\mathbf{R}_{\mathbf{o}_i}^{-1}(\mathbf{p}_j - \mathbf{p}_i), \arccos \mathbf{o}_i^\mathsf{T} \mathbf{o}_j), \tag{8}$$

$$\mathbb{R}^3 \times S^2: \quad [(\mathbf{p}_i, \mathbf{o}_i), (\mathbf{p}_j, \mathbf{o}_j)] \;\mapsto\; a_{ij} = \begin{pmatrix} \mathbf{o}_i^\mathsf{T}(\mathbf{p}_j - \mathbf{p}_i) \\ \|(\mathbf{p}_j - \mathbf{p}_i) - \mathbf{o}_i^\mathsf{T}(\mathbf{p}_j - \mathbf{p}_i)\mathbf{o}_i\| \\ \arccos \mathbf{o}_i^\mathsf{T} \mathbf{o}_j \end{pmatrix}, \tag{9}$$

$$\mathrm{SE}(3): \quad [(\mathbf{p}_i, \mathbf{R}_i), (\mathbf{p}_j, \mathbf{R}_j)] \;\mapsto\; a_{ij} = (\mathbf{R}_i^{-1}(\mathbf{p}_j - \mathbf{p}_i), \mathbf{R}_i^{-1}\mathbf{R}_j). \tag{10}$$

*Proof.* A proof is provided in the appendix, section B.2. $\qquad\square$

The three components of $a_{ij}$ in Eqs. 8, 9 share the same interpretation: the first two decompose displacement $\mathbf{p}_j - \mathbf{p}_i$ into shifts parallel and orthogonal to $\mathbf{o}_i$, the third represents the angle between $\mathbf{o}_i$ and $\mathbf{o}_j$. As for any coordinate system, these mappings are not unique: we can rewrite Eq. 9

in polar coordinates as $a_{ij} = (\|\mathbf{p}_j - \mathbf{p}_i\|, \arccos \mathbf{o}_i^\intercal (\mathbf{p}_j - \mathbf{p}_i), \arccos \mathbf{o}_i^\intercal \mathbf{o}_j)^\intercal$: the invariants used in DimeNet (Gasteiger et al., 2019). Eqs. 8-10 implement $g_i^{-1} g_j$. The following is proven in Appx. B.4

**Corollary 1.1.** *Message passing networks in geometric graphs over $\mathbb{R}^n \times S^{n-1}$ and $SE(n)$, with message functions conditioned on the attributes of Thm. 1, are equivariant universal approximators.*

## 3.2 SEPARABLE GROUP CONVOLUTION IN POSITION-ORIENTATION SPACE

Based on the bijective equivalence class embedding from Sec. 3.1, we define a group convolution as

$$f^{out}(\mathbf{p}, \mathbf{o}) = \int_{\mathbb{R}^3} \int_{S^2} k([(\mathbf{p}, \mathbf{o}), (\mathbf{p}', \mathbf{o}')]) f(\mathbf{p}', \mathbf{o}') \mathrm{d}\mathbf{p}' \mathrm{d}\mathbf{o}'. \tag{11}$$

Here, $f : \mathbb{R}^3 \times S^2 \to \mathbb{R}^{C_{in}}$ represents the input feature map with $C_{in}$ channels. the kernel $k([(\mathbf{p}, \mathbf{o}), (\mathbf{p}', \mathbf{o}')]) \in \mathbb{R}^{C_{out} \times C_{in}}$ returns a linear transformation matrix for each equivalence class of point pairs, resulting in an output signal $f^{out} : \mathbb{R}^3 \times S^2 \to \mathbb{R}^{C_{out}}$ with $C_{out}$ channels.

We use the fact that the third component of the invariant attribute in Eq. 9 only depends on orientations to separate the $\mathbb{R}^3 \times S^2$ convolution into three parts: spatial convolution, spherical convolution, and channel mixing, as illustrated in figure 1. This approach combines the efficiency strategies of Knigge et al. (2022); Chollet (2017), to separate group convolutions over different group parts, and channel interactions from spatial interactions –generally homogeneous space interactions–, respectively. Following this strategy, we factorize the convolution kernel into three parts:

$$k([(\mathbf{p}, \mathbf{o}), (\mathbf{p}', \mathbf{o}')]) = K^{(3)} \, k^{(2)}(\mathbf{o}^\intercal \mathbf{o}') \, k^{(1)}(\mathbf{o}^\intercal (\mathbf{p}' - \mathbf{p}), \|\mathbf{o} \perp (\mathbf{p}' - \mathbf{p})\|), \tag{12}$$

with $k^{(1)} : \mathbb{R} \times \mathbb{R}_{\geq 0} \to \mathbb{R}^{C_{in}}$ the channel-wise spatial mixing kernel, $k^{(2)} : [0, \pi] \to \mathbb{R}^{C_{in}}$ the channel-wise orientation mixing kernel, and $K^{(3)} \in \mathbb{R}^{C_{out} \times C_{in}}$ the channel mixing kernel. Substituting the factorized kernel in Eq. 11 allows us to decompose it into three distinct modules:

$$f \longrightarrow \boxed{\texttt{SpatialGConv}} \longrightarrow \boxed{\texttt{SphericalGConv}} \longrightarrow \boxed{\texttt{Linear}} \longrightarrow f^{out}, \tag{13}$$

with $\boxed{\texttt{Linear}}$ the usual linear layer –implementing $K^{(3)}$– and

$$\boxed{\texttt{SpatialGConv}}: \qquad f_{i,o}^{(1)} = \sum_{j \in \mathcal{N}(i)} k^{(1)}(\mathbf{o}_o^\intercal (\mathbf{p}'_j - \mathbf{p}_i), \|\mathbf{o}_o \perp (\mathbf{p}_j - \mathbf{p}_i)\|) \odot f_{j,o} \tag{14}$$

$$\boxed{\texttt{SphericalGConv}}: \qquad f_{i,o}^{(2)} = \sum_{o'=0}^{N-1} k^{(2)}(\mathbf{o}_o^\intercal \mathbf{o}'_o) \odot f_{i,o'}^{(1)}. \tag{15}$$

Here $\odot$ denotes element-wise multiplication and we turned the integrals of Eq. 11 into discrete sums via Eq. 5. In the discrete implementations, the position-orientation space feature maps $f$ are of dimension $[P, N, C]$, with $P$ the number of points in the point cloud, $N$ the number of orientations on a precomputed spherical grid that is shared over all positions, and with $C$ the number of channels. We use $f_{j,o}$ to denote the feature vector at position $\mathbf{p}_j$ and orientation $\mathbf{o}_o$. Unlike traditional point-cloud methods, we introduce an additional axis to index features over different orientations.

Each of the operations in Eq. 13 is highly efficient. The most expensive module is SpatialGConv, as it requires aggregating over spatial neighborhoods. We minimize the computation cost of this operation by avoiding channel mixing in this step, essentially performing a depth-wise separable convolution (Chollet, 2017). SphericalGConv is efficient as it can be batched over positions and channels. Moreover, since the spherical grids are shared over all positions, the kernel $k^{(2)}$ can be precomputed, resulting in a kernel of shape $[N, N', C]$. Eq. 15 is then implemented as an `einsum` over the $N'$ axis of the kernel and input with shape $[P, N', C]$.

While the feature maps in Eq. 14, 15 are discretized, the convolution kernels are still continuous. We parameterize these using Multilayer Perceptrons (MLPs) that take attributes as input. Following standard practices, we first embed the attributes, which we choose to do using a Polynomial Embedding (PE) (cf. Appx. C). That is, we sample the kernel via $a \to \boxed{\texttt{PE}} \to \boxed{\texttt{MLP}} \to k$.

## 3.3 THE PØNITA ARCHITECTURE

We use the separable $\mathbb{R}^3 \times S^2$ group convolution in a `ConvNeXt` (Liu et al., 2022b) layer structure:

$$f \underset{\downarrow}{\to} \boxed{\texttt{SpatialGConv}} \to \boxed{\texttt{SphericalGConv}} \to \boxed{\texttt{LayerNorm}} \to \boxed{\texttt{Linear}} \to \boxed{\texttt{Activation}} \to \boxed{\texttt{Linear}} \underset{\uparrow}{\to} f^{out}$$

and use it to construct a simple fully convolutional neural network given by:

$$f \rightarrow \boxed{\texttt{NodeEmbed}} \rightarrow \boxed{\texttt{ConvNeXt} \times L} \rightarrow \boxed{\texttt{Readout}} \rightarrow f^{out} \quad . \tag{16}$$

We call this position-orientation space network based on invariant attributes PΘNITA. Since PΘNITA uses position-orientation space feature maps, we require a method to embed inputs onto spherical signals, and readout outputs from it. We do so via the following node-wise modules for embedding and predicting scalar fields $s : \mathcal{V} \rightarrow \mathbb{R}$ and vector fields $\mathbf{v} : \mathcal{V} \rightarrow \mathbb{R}^3$:

$$\boxed{\texttt{SphereToScalar}} : \; s_i = \sum_{o=0}^{N-1} f_{i,o}, \qquad \boxed{\texttt{ScalarToSphere}} : \; f_{i,o} = s_i,$$

$$\boxed{\texttt{SphereToVec}} : \qquad \mathbf{v}_i = \sum_{o=0}^{N-1} f_{i,o}\, \mathbf{o}_o\,, \quad \boxed{\texttt{VecToSphere}} : \qquad f_{i,o} = \mathbf{v}_i^\mathsf{T} \mathbf{o}_o\,.$$

The VecToSphere module is essentially a spherical harmonic embedding of frequency one, followed by an inverse spherical Fourier transform: $f_i(\mathbf{o}) = \mathcal{F}_{S^2}^{-1}[Y^{(l=1)}(\mathbf{v})](\mathbf{o})$. This perspective offers a generalized approach for embedding higher-order tensors onto the sphere by using them as coefficients in an inverse Fourier transform, e.g., using the `e3nn` library (Geiger & Smidt, 2022).

## 4 RELATED WORK

The theoretical part of our work is inspired by other works on the construction of equivariant neural networks on homogeneous spaces, such as (Cohen et al., 2019; Kondor & Trivedi, 2018; Bekkers, 2019). Each of these papers contains a version of the 'convolution is all you need' theorem, which we take as a motivating starting point in this paper (cf. section 2.2). For a comprehensive unifying coverage of equivariant methods, we recommend (Weiler et al., 2023).

Group convolutions are typically defined in *regular* or *steerable / tensor field* form. The regular group convolution viewpoint -which we adopt in this paper- naturally extends convolution as template matching of a kernel over an underlying signal, but now matching all transformation in a group $G$ instead of translation only (Cohen & Welling, 2016; Bekkers, 2019). On gridded data, such as images, regular group convolutions either achieve *exact equivariance* by considering symmetry groups of the grid (Cohen & Welling, 2016; Worrall & Brostow, 2018), or *numerically approximate equivariance* by relying on interpolation (Bekkers et al., 2018; Kuipers & Bekkers, 2023), basis functions (Weiler et al., 2018; Sosnovik et al., 2019), or continuous MLP-based kernels (Finzi et al., 2020a; Knigge et al., 2022). The steerable approach bypasses the need for discretizations by working fields of vectors that transform under (irreducible) representations of the $\mathrm{SO}(n)$ group Weiler et al. (2021). Under the 'convolution is all you need' paradigm, regular and steerable methods are equivalent and they relate to each other via $\mathrm{SO}(n)$ Fourier transforms (Kondor et al., 2018; Cesa et al., 2021). In Appx. A we make precise how to interpret PΘNITA in either the steerable or regular form.

In the field of deep learning-based molecular modeling, in which physical modeling based on Spherical Harmonics, Clebsch-Gordan tensor products, and representation theory of $\mathrm{SO}(n)$ is more common, there is a tendency to rely on the tensor field paradigm. Influential contributions in this field are *Tensor Field Network* (Thomas et al., 2018) and *Cormorant* (Anderson et al., 2019). Limitations of such methods are the high computational overhead due to the specialized Clebsch-Gordan tensor products and the need for a solid understanding of representation theory. An advantage of the tensor field approach is that one obtains exact equivariance, whereas the regular group convolution approach requires discretizations of the group, which leads to equivariance only up to the discretization resolution. In this work, however, we show that these numerical errors are not detrimental to performance and opt for simplicity and familiarity with the standard convolutional paradigm.

Another class of equivariant methods is based on message passing with invariant geometric attributes, such as DimeNet (Gasteiger et al., 2019), GemNet (Gasteiger et al., 2021), and SphereNet (Liu et al., 2022a). These are closest to the regular group convolution paradigm and could be framed as non-linear group convolutions (Brandstetter et al., 2021) over the *homogeneous space of positions and orientations*. Our method is different in that it is based on the standard convolution paradigm instead of intricate wiring of interaction modules, and we use dense grids over the sphere instead of sparse equivariant grids defined by edge directions. A notable result from (Gasteiger et al., 2021) is that networks over position-orientation space are *universal equivariant approximators* and thus as expressive as equivariant methods that use feature representations over the full group $\mathrm{SE}(n)$.

Our method connects to a line of papers on theory and algorithms for the processing of functions over position-orientation space via *orientation scores*. See e.g., (Duits & Franken, 2010a;b) and (Janssen et al., 2018) for respectively 2D and 3D applications (Duits et al., 2021). Noteworthy, Portegies et al. (2015) derive invariant attributes for position-orientation pairs derived via the logarithmic map on SE(3). In particular, these works address the processing of Diffusion-weighted MRI data -which are signals over $\mathbb{R}^3 \times S^2$, for which our work provides a simple equivariant learning recipe.

## 5 EXPERIMENTS

In this section, we evaluate our approach. Comprehensive implementation details, including architecture specifications and optimization techniques, can be found in Appx. C and D.

**Benchmark 1: Predicting interatomic potential energy and forces on rMD17.** rMD17 (Christensen & Von Lilienfeld, 2020) is a dataset comprising molecular dynamics trajectories of ten small molecules. Each molecule is represented as an atomic point cloud consisting of 3D atom positions and atomic numbers. The objective is to predict the energy-conserving force field for each molecule. The regression targets are the total energy of each molecule and the force on each atom. Following common practice and the laws of physics, we predict the energy with an invariant model and compute the forces at each atom as the gradients of the predicted energy with respect to the positions of atoms. We utilize the PΘNITA model as given in Eq . 16, with a linear embedding layer that takes one-hot encodings of the atom numbers as input and maps them onto the sphere via the ScalarToSphere module. Following MACE (Batatia et al., 2022b), DimeNet and GemNet (Gasteiger et al., 2019; 2021), we predict a node-wise energy $E_{io}^l$ after every layer $l$ using a linear readout layer and obtain the total energy as the sum of all predicted energies $E = \sum_{i,o,l} E_{io}^l$.

Our results are reported in Tab. 1. As a baseline, we construct the same architecture as PΘNITA , except that internal feature maps live solely in position space $\mathbb{R}^3$. We label it PNITA . In this case, the pairwise attribute is simply the distance (Eq. 7), which makes PNITA similar to SchNet (Schütt et al., 2023). The inability of such models to learn direction-sensitive representations explains the large performance gap relative to PΘNITA and other methods that capture directional information.

Tab. 1 also shows that PΘNITA either matches or outperforms the state-of-the-art on several molecules while being remarkably faster. PΘNITA is about 3.5 times faster than the established NequIP method (Batzner et al., 2022): a type of steerable tensor field network (Thomas et al., 2018). We benchmarked their optimized codebase within our training framework and recorded a training time per epoch of 20.7, compared to 5.7 seconds for PNITA and 6.1 for PΘNITA . Additionally, PΘNITA stands out for its *simplicity*. It is a general purpose convolutional network, whose design does not require in-depth knowledge of representation theory and molecular modeling like UNiTE (Qiao et al., 2022), MACE (Batatia et al., 2022b), NequIP (Batzner et al., 2022), Allegro (Musaelian et al., 2023) and BOTNet (Batatia et al., 2022a), nor does it involve an intricate wiring of interaction layers as in GemNet (Gasteiger et al., 2021) and ViSNet (Wang et al., 2022).

**Benchmark 2: Generating molecules with equivariant diffusion models trained on QM9.** Diffusion models have proven to be very effective for the generation of data such as images (Song et al., 2021) and molecules (Hoogeboom et al., 2022). Prior work Gebauer et al. (2019); Simonovsky & Komodakis (2018); Simm et al. (2021) has demonstrated the importance of leveraging molecular symmetries for generalization. Through weight-sharing, PΘNITA is designed to represent such symmetries, making it a promising candidate for this application. Recently advancements like EDMs (Hoogeboom et al., 2022), and MiDi (Vignac et al., 2023) have employed denoising diffusion for molecular data generation. These models utilize EGNN (Satorras et al., 2021) as an equivariant denoising diffusion model, which operates on both atomic positions and types.

We extend EDMs by building a similar architecture for molecular generation, incorporating a joint diffusion process for both continuous coordinates and discrete features. However, we substitute the EGNN backbone with our proposed method PΘNITA . For the readout layer in Eq. 16, we use the Sphere2Vec module to predict denoising displacement vectors. We train on the QM9 dataset (Ramakrishnan R., 2014), a standard dataset containing molecular properties, one-hot representations of atom types $(\mathrm{H, C, N, O, F})$ and 3D coordinates for 130K molecules with up to 9 heavy atoms.

Tab. 2 reveals that PΘNITA outperforms several baselines in terms of atom stability –determined by atoms with the right valency–, and molecule stability –determined by the proportion of molecules for which all atoms are stable–. We attribute this significant improvement to PΘNITA 's ability to

Table 1: Mean absolute errors (MAE) of energy (E) (kcal/mol) and force (F) (kcal/mol/Å) for 10 small organic molecules on rMD17 compared with the state-of-the-art.

| Molecule | | UNiTE | GemNet | NequIP (20.7 sec/epoch) | BOTNet | Allegro | MACE | ViSNet | PNITA (5.7 sec/epoch) | PΘNITA (6.1 sec/epoch) |
|---|---|---|---|---|---|---|---|---|---|---|
| Aspirin | E | 2.4 | - | 2.3 | 2.3 | 2.3 | 2.2 | 1.9 | 4.7 | **1.7**$_{\pm 0.03}$ |
| | F | 7.6 | 9.5 | 8.2 | 8.5 | 7.3 | 6.6 | 6.6 | 16.3 | **5.8**$_{\pm 0.18}$ |
| Azobenzene | E | 1.1 | - | 0.7 | 0.7 | 1.2 | 1.2 | **0.7** | 3.2 | **0.7**$_{\pm 0.01}$ |
| | F | 4.2 | - | 2.9 | 3.3 | 2.6 | 3.0 | 2.5 | 12.2 | **2.3**$_{\pm 0.11}$ |
| Benzene | E | 0.07 | - | 0.04 | **0.03** | 0.3 | 0.4 | **0.03** | 0.2 | 0.17$_{\pm 0.01}$ |
| | F | 0.73 | 0.5 | 0.3 | 0.3 | **0.2** | 0.3 | 0.2 | 0.4 | 0.3$_{\pm 0.00}$ |
| Ethanol | E | 0.62 | - | 0.4 | 0.4 | 0.4 | 0.4 | **0.3** | 0.7 | 0.4$_{\pm 0.02}$ |
| | F | 3.7 | 3.6 | 2.8 | 3.2 | **2.1** | **2.1** | 2.3 | 4.1 | 2.5$_{\pm 0.09}$ |
| Malonaldehyde | E | 1.1 | - | 0.8 | 0.8 | 0.6 | 0.8 | **0.6** | 0.9 | **0.6**$_{\pm 0.05}$ |
| | F | 6.6 | 6.6 | 5.1 | 5.8 | **3.6** | 4.1 | 3.9 | 5.1 | 4.0$_{\pm 0.20}$ |
| Naphthalene | E | 0.46 | - | **0.2** | **0.2** | **0.2** | 0.4 | 0.2 | 1.1 | 0.3$_{\pm 0.00}$ |
| | F | 2.6 | 1.9 | 1.3 | 1.8 | **0.9** | 1.6 | 1.3 | 5.6 | 1.3$_{\pm 0.00}$ |
| Paracetamol | E | 1.9 | - | 1.4 | 1.3 | 1.5 | 1.3 | **1.1** | 2.8 | **1.1**$_{\pm 0.03}$ |
| | F | 7.1 | - | 5.9 | 5.8 | 4.9 | 4.8 | 4.5 | 11.4 | **4.3**$_{\pm 0.22}$ |
| Salicylic acid | E | 0.73 | - | 0.7 | 0.8 | 0.9 | 0.9 | **0.7** | 1.7 | **0.7**$_{\pm 0.00}$ |
| | F | 3.8 | 5.3 | 4.0 | 4.3 | **2.9** | 3.1 | 3.4 | 8.6 | 3.3$_{\pm 0.07}$ |
| Toluene | E | 0.45 | - | 0.3 | 0.3 | 0.4 | 0.5 | **0.3** | 0.6 | **0.3**$_{\pm 0.01}$ |
| | F | 2.5 | 2.2 | 1.6 | 1.9 | 1.8 | 1.5 | **1.1** | 3.4 | 1.3$_{\pm 0.07}$ |
| Uracil | E | 0.58 | - | 0.4 | 0.4 | 0.6 | 0.5 | **0.3** | 0.9 | 0.4$_{\pm 0.04}$ |
| | F | 3.8 | 3.8 | 3.1 | 3.2 | **1.8** | 2.1 | 2.1 | 5.6 | 2.4$_{\pm 0.09}$ |

Table 2: Molecule generation via denoising diffusion models trained on QM9. Negative Log Likelihood (NLL), atom, and molecule stability.

| Models | NLL | Atom stability | Mol stability |
|---|---|---|---|
| E-NF* | -59.7 | 85.0 | 4.9 |
| G-Schnet* | N.A. | 95.7 | 68.1 |
| GDM* | -94.7 | 97.0 | 63.2 |
| GDM-aug* | -92.5 | 97.6 | 71.6 |
| EDM* | $-110.7_{\pm 1.5}$ | 98.7$_{\pm 0.1}$ | 82.0$_{\pm 0.4}$ |
| **PΘNITA** | **−137.4** | **98.9** | **87.8** |

Table 3: Mean squared error on $N$-body trajectory prediction.

| Method | MSE | sec/epoch |
|---|---|---|
| SE(3)-Tr. | .0244 | |
| TFN | .0155 | |
| NMP | .0107 | |
| Radial Field | .0104 | |
| EGNN | .0070$_{\pm .00022}$ | |
| SEGNN$_{G+P}$ | .0043$_{\pm .00015}$ | 1.59 |
| **CGENN** | **.0039**$_{\pm .0001}$ | |
| PΘNITA | .0043$_{\pm .0001}$ | 0.66 |

*learn orientation-sensitive representations*: a feature absent in other equivariant diffusion models such as G-Schnet (Gebauer et al., 2019), E-NF (Garcia Satorras et al., 2021) and therein introduced non-equivariant baseline Graph Diffusion models (GDM) trained with and without augmentation.

**Benchmark 3: Predicting trajectories in N-body systems.** Finally, we benchmark PΘNITA on the charged N-body particle system experiment proposed in Kipf et al. (2018). It involves five particles that carry either a positive or negative charge, each having initial position and velocity in a 3D space. The objective is to predict all particle positions after a fixed number of time steps. We adapt the experimental setup from (Satorras et al., 2021; Brandstetter et al., 2021) and use PΘNITA for trajectory prediction. Following the baseline methods, PΘNITA takes two scalars: velocity magnitude and charge, and two vectors: initial velocity and direction to the center of the system, as input, and outputs a displacement vector. We measure the performance in Mean Squared Error (MSE) and compare against SE(3)-Transformers (Fuchs et al., 2020), Tensor Field Networks (Thomas et al., 2018), Neural Massage Passing (Gilmer et al., 2017b), Radial Field (Köhler et al., 2020), EGNN (Satorras et al., 2021), SEGNN (Brandstetter et al., 2021) and CGENN (Ruhe et al., 2023). Tab. 3 shows the results and the reproduced train-time/epoch for SEGNN, showing that PΘNITA is $\pm 2.5$ times faster. Once again, our versatile equivariant architecture, PΘNITA , delivers outstanding performance.

**Benchmark 4: Superpixel MNIST** and **Benchmark 5: QM9 regression**. In App. E.1 and E.2 we further show the potency of PΘNITA with state-of-the-art results on benchmarks in 2D and 3D and further compare an edge-index-induced $\mathbb{R}^3 \times S^2$ point cloud vs a grid-based (fiber bundle) approach.

## 6 CONCLUSION

In summary, our method simplifies the construction of equivariant neural networks by following the paradigm of weight-sharing, achieving state-of-the-art results in various benchmarks. In Thm. 1, we derived geometric attributes that fully characterize the space of equivalence classes of point-pairs in a homogeneous space of $SE(n)$. As such, *these attributes are all you need* to facilitate $SE(n)$ equivariant weight-sharing. As an application of the theory, we propose PΘNITA : an efficient convolutional architecture for equivariant processing of 3D point clouds. We show that our general-purpose equivariant approach serves as a compelling alternative to prevailing tensor field methods.

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

## A  THE FIBER BUNDLE VS POINT CLOUD VIEWPOINT

### A.1  REGULAR VS STEERABLE GROUP CONVOLUTIONS

In the field of group equivariant deep learning, one often adopts a steerable or regular convolutional viewpoint on equivariant neural networks (Weiler & Cesa, 2019). Let us consider the case of $SE(n)$ equivariance, and note that the group $SE(n)$ is a semi-direct product between the translation group and the rotation group $SO(n)$. We then consider functions (feature fields) over domains $X$ that are homogeneous spaces of $SE(n)$, namely the case $X = \mathbb{R}^n$, $X = \mathbb{R}^n \times S^{n-1}$, and $X = SE(n)$. The difference between steerable and regular group convolutions then lies in the type of feature fields that are considered.

With *regular group convolutions* one considers multi-channel scalar fields $f : X \to \mathbb{R}^c$ of the homogeneous space $X$. And convolutions are of the form

$$f^{out}(y) = \int_X k(g_y^{-1} x) f^{in}(x) \mathrm{d}x \,, \tag{A1}$$

with $g_y$ a representative of the point $y$ such that $y = g_y x_0$ with $x_0$ some arbitrary origin in $X$. Regular feature fields, and convolution kernels, transform via the regular representation of $SE(n)$ via

$$(\rho^L(g)k)(x) := k(g^{-1}x) \,.$$

Given this, regular group convolutions can be simply considered as template matching between a roto-translated convolution kernel (by transforming it with the regular representation) and the underlying signal, where similarity is measured by the $\mathbb{L}_2$ inner product. When $X \equiv SE(n)/H$ with $H$ a non-trivial group, such as $H = SO(3)$ when $X = \mathbb{R}^n$ or $H = SO(2)$ when $X = \mathbb{R}^3 \times S^2$, then the kernel $k$ should satisfy a kernel constraint $\forall_{h \in H} : k(x) = k(hx)$, see (Bekkers, 2019). As discussed in the main body of this paper, this constraint is automatically satisfied if we parametrize the kernel by the invariant attributes $a_{ij}$ of Theorem 1. The advantage of regular group convolutions is that the codomain of the feature fields are just scalars and one can use any point-wise activation function without breaking equivariance. A drawback of such methods though is that one needs to discretize the domain $X$, including the fibers $S^{n-1}$ or $SO(n)$, which leads to numerical inexactness of the equivariance property[1].

With *steerable group convolutions* one considers *feature fields* $f : \mathbb{R}^n \to V_\rho$ over a base space $\mathbb{R}^n$ with a codomain which is a vector space $V_\rho$ on which a representation $\rho$ of $SO(n)$ can act. Such feature fields transform via the so-called *induced representation* of $SE(n)$, defined as

$$([\mathrm{Ind}_{SO(n)}^{SE(n)} \rho](g)f)(\mathbf{x}) = \rho(\mathbf{R}) \, f(g^{-1}\mathbf{x}) \,.$$

I.e., the induced representation $[\mathrm{Ind}_{SO(n)}^{SE(n)} \rho]$ does not only transform the feature map's domain but also its codomain via the representation $\rho$. For example, if one roto-translates a vector field $\mathbf{v} : \mathbb{R}^3 \to \mathbb{R}^3$, the vectors $\mathbf{v}(\mathbf{x})$ in the field at location $\mathbf{x}$ should be moved to the new locations $g^{-1}\mathbf{x}$, but also their values should be rotated via $\mathbf{R}\mathbf{v}(g^{-1}\mathbf{x})$. The steerable convolutions are then simply given by

$$f^{out}(\mathbf{y}) = \int_{\mathbb{R}^n} k(\mathbf{x} - \mathbf{y}) f^{in}(\mathbf{x}) \mathrm{d}\mathbf{x} \,. \tag{A2}$$

Now, however, the kernel needs to satisfy the kernel constraint

$$k(\mathbf{R}\,\mathbf{x}) = \rho_{out}(\mathbf{R})k(\mathbf{x})\rho_{in}(\mathbf{R}^{-1}) \,,$$

with $\rho_{in}$ and $\rho_{out}$ the representations that define the vectors spaces of the codomains of the input and output feature fields respectively, see e.g. (Weiler & Cesa, 2019; Cesa et al., 2021). An advantage of the steerable approach is that one can obtain exact equivariance by choosing to let the vector spaces $V_\rho$ transform via representations $\rho = \oplus_l \rho_l$ that are given as a direct sum of irreducible representations. Such irreducible representations do not require a grid and rotations can be computed exactly. A drawback of such an approach is, however, that one can no longer apply arbitrary point-wise activation functions but one needs specialized operations on $V_\rho$ in order not to break equivariance. This limits expressivity (Weiler & Cesa, 2019). Another drawback, arguably, is that this approach requires considerable understanding of representation theory for proper use in practice.

---

[1]Our experiments, however, show that this approximate equivariance is not detrimental to performance

## A.2 Regular Group Convolutions implemented as Steerable Group Convolutions

It is important to note that in the regular group convolution case for $SE(n)$ equivariance, one recognizes a semi-direct product structure of the group $SE(n) = (\mathbb{R}^n, +) \rtimes SO(n)$, with $(\mathbb{R}^n, +)$ the translation group, which as a space can be identified with position space $\mathbb{R}^n$. When one then considers regular group convolutions over $X = \mathbb{R}^n \times S^{n-1}$ or $X = \mathbb{R}^n \times SO(n)$, one could think of the feature fields $f : \mathbb{R}^n \times Y \to \mathbb{R}^c$ as assigning to every position $\mathbf{x} \in \mathbb{R}^n$ a function $f_{\mathbf{x}} := f(\mathbf{x}, \cdot)$ over the space $Y$, with here $Y = S^{n-1}$ or $Y = SO(n)$. These functions $f_{\mathbf{x}}$ are to be considered vectors in the infinite-dimensional vector space of square-integrable functions $\mathbb{L}_2(Y)$. For this vector space we know the left-regular representations $\rho_{\mathbb{L}_2(Y)}^L$, which are simply given as

$$(\rho_{\mathbb{L}_2(S^{n-1})}^L(\mathbf{R})f_{\mathbf{x}})(\mathbf{o}) = f_{\mathbf{x}}(\mathbf{R}^T\mathbf{o}), \quad \text{and} \quad (\rho_{\mathbb{L}_2(SO(n))}^L(\mathbf{R})f_{\mathbf{x}})(\mathbf{R}') = f_{\mathbf{x}}(\mathbf{R}^T\mathbf{R}').$$

Thus, one can think of regular group convolutions as working with feature fields $f : \mathbb{R}^n \to V_{\rho_{\mathbb{L}_2(Y)}^L}$.

Further, note that any $SO(n)$ representation can be decomposed into a direct sum of irreducible representations via a change of basis. I.e., there is a linear change of basis such that $V_{\rho_{\mathbb{L}_2(Y)}^L} \equiv V_\rho$ for some $\rho = \oplus_l \rho_l$, with $\rho_l$ an irreducible representation. This change of basis is given by the Fourier transform on $Y$. Namely, a function $f_{\mathbf{x}} \in V_{\rho_{\mathbb{L}_2(Y)}^L}$ can be transformed to a vector of Fourier coefficients via $\hat{\mathbf{f}}_{\mathbf{x}} = \mathcal{F}_Y[f_{\mathbf{x}}] \in V_\rho$, and back via the inverse Fourier transform $f_{\mathbf{x}} = \mathcal{F}_Y^{-1}[\hat{\mathbf{f}}_{\mathbf{x}}]$.

Regular group convolutions can thus either be implemented as steerable convolutions with fields of signals over $Y$, i.e., $f : \mathbb{R}^n \to V_{\rho_{\mathbb{L}_2(Y)}^L}$, or via fields of irreducible representations $f : \mathbb{R}^n \to V_\rho$ that can be thought of as fields of Fourier coefficients. In this latter case, the kernel constraint is satisfied by parametrizing $k$ using the Clebsch-Gordan tensor product (Cesa et al., 2021; Brandstetter et al., 2021). This approach is in fact how group equivariant neural networks are often implemented (Weiler & Cesa, 2019; Thomas et al., 2018), however, it has the drawback that one either has to use specialized activation functions or use activation functions of the form `InverseFourier → ActivationFunction → FourierTransform`, which leads to computational overhead and leads to discretization artifacts that break exact equivariance. In this paper, we present a fully *regular* group convolution viewpoint that does not require Fourier transforms or specialized activation functions.

## A.3 Fiber Bundle Viewpoint for PΘNITA

From a fiber bundle-theoretic viewpoint, the feature fields $f : \mathbb{R}^n \to V_\rho$ can be regarded as sections of vector bundles associated with the principal fiber bundle $\mathbb{R}^n \times SO(n) \to \mathbb{R}^n$ (Weiler et al., 2023; Aronsson, 2023). As such, without going into further technical details behind this theory, we merely use this connection to refer to the vector spaces $V_\rho$ as fibers. The PΘNITA model processes feature fields $f : \mathbb{R}^n \to V_{\rho_{\mathbb{L}_2(S^{n-1})}^L}$ with vector spaces of spherical signals $V_{\rho_{\mathbb{L}_2(S^{n-1})}^L}$ as fibers. Instead of describing these signals in a basis of irreducible representations (the Fourier approach), we sample them on a grid $\mathcal{S} \subset S^{n-1}$ of orientations. See Figure 1 in which the grid is depicted as a set of yellow dots, and the actual spherical signal is depicted as a gray-scale density. Our model makes efficient use of the fiber bundle structure by separating convolutional operations over the field into a spatial interaction part (convolving over $\mathbb{R}^n$) followed by a within-fiber orientation interaction part (convolving over $\mathbb{S}^{n-1}$).

## A.4 A Point Cloud Viewpoint for PΘNITA

The efficient implementation of PΘNITA is made possible by the fact that each node shares the same grid (fiber) of orientations $\mathcal{S}$. Let us denote with $\mathcal{S}_{\mathbf{x}} \subset S^{n-1}$ the grid of orientations available at position $\mathbf{x}$. If we share the same grid over all positions, i.e., $\mathcal{S}_{\mathbf{x}} = \mathcal{S}$, we can decide to spatially convolve per orientation in the grid as follows

$$f^{out}(\mathbf{x}, \mathbf{o}) = \int_{\mathbb{R}^n} k([(\mathbf{x}, \mathbf{o}), (\mathbf{x}', \mathbf{o})]) f^{in}(\mathbf{x}', \mathbf{o}) \mathrm{d}\mathbf{x}',$$

which is the continuous formulation of the discretized spatial convolution of Eq. (14). This purely spatial convolution is made possible by the fact that at each position $\mathbf{x}'$ we can sample the same

orientation $\mathbf{o} \in \mathcal{S}$. However, if we do not share the same grid $\mathcal{S}$ over all positions we can no longer separate the convolution. In this case, we adopt a point-cloud viewpoint.

There are natural instances in which one has to compute with point clouds in position-orientation space. Examples include triangulated shape meshes, which can be represented by the centroids of the triangles $\mathbf{p}$ and their normal vectors $\mathbf{o}$. See e.g. (De Haan et al., 2020). In contrast to (De Haan et al., 2020) our method would take the curvature of the surface mesh into account (via the invariant $\mathbf{o}_i^T \mathbf{o}_j$), as well as out off plane distance, however, the method by De Haan et al. (2020) takes anisotropic features within the plane into account whereas the second invariant in Eq. (9)) restricts the kernel to be isotropic within the tangent plane.

Another instance of a position-orientation space point cloud would be in a message passing over the edges in an atomic point cloud, as in DimeNet/GemNet (Gasteiger et al., 2019; 2021). In those works, a grid is assigned to each atom location $\mathbf{x}_i$ based on the direction towards neighboring atom locations $\mathbf{x}_j$ that are connected by covalent bonds. I.e., each atom obtains a grid $\mathcal{S}_{\mathbf{x}_i} = \{ \frac{\mathbf{x}_j - \mathbf{x}_i}{\|\mathbf{x}_j - \mathbf{x}_i\|} \mid j \in \mathcal{N}(i)\}$, with $\mathcal{N}(i)$ the set of neighbor indices of node $i$.

Convolutions in these case can no longer be separable and include spatial and orientation interactions simultaneously:

$$f^{out}(\mathbf{x}_i, \mathbf{o}_i) = \sum_{j \in \mathcal{N}(i)} \sum_{\mathbf{o} \in \mathcal{S}_{\mathbf{x}_j}} k([(\mathbf{x}_i, \mathbf{o}_i), (\mathbf{x}_j, \mathbf{o})]) f^{in}(\mathbf{x}_j, \mathbf{o}) \,.$$

This is the discretized version of the full continuous convolution of Eq. (11).

## A.5 Implementation

Both versions of position-orientation space convolutions –the fiber bundle approach as well as the point cloud approach– are made available in the public repository `https://github.com/ebekkers/ponita` .

## B Proofs

### B.1 Proof of Lemma 1

Lemma 1 shows that equivalence classes of point pairs $[x_i, x_j] = \{(g\,x_i, g\,x_j) \mid g \in G)\}$ correspond to orbits $H\,g_i^{-1} x_j$, for any chosen $g_i$ such that $x_i = g_i\,x_0$, with $x_i, x_j, x_0 \in X$. In the following proof, one might recognize that the space of equivalence classes $X \times X/\sim$ can also be identified with the space $(H, H)$-double cosets in $G$. That is $X \times X/\sim \equiv H\backslash G/H \equiv H\backslash X$. The proof of Lemma 1 is as follows:

*Proof.* To prove the bijection given by equation 6 we rewrite the equivalence class into the form of a double coset as follows. For any representatives $(g_i, g_j)$ we have

$$
\begin{aligned}
[x_i, x_j] \quad &= \quad &&\{(g\,x_i, g\,x_j) \mid g \in G\} \\
&\overset{\forall_{h \in H}: h\,x_0 = x_0}{=} &&\{(g\,g_i\,h_i\,m_0, g\,g_j\,h_j\,m_0) \mid g \in G, h_i, h_j \in H\} \\
&\overset{g \to g\,h_i^{-1} g_i^{-1}}{=} &&\{(h_i^{-1} g_i^{-1} \cancel{g}\cancel{g}\,g_j\,h_j\,m_0, g\,m_0) \mid g \in G, h_i, h_j \in H\} \\
&\overset{g^{-1}g=e}{=} &&\{(h_i^{-1} g_i^{-1} g_j\,h_j\,m_0, G\,m_0) \mid h_i, h_j \in H\} \\
&\overset{x_j = g_j\,m_0,\ h_j\,x_0 = m_0}{=} &&\{(h_i^{-1} g_i^{-1} x_j, G\,m_0) \mid h_i, h_j \in H\} \\
&\overset{G\,m_0 = M}{\Leftrightarrow} &&\{h_i\,g_i^{-1} x_j \mid h_i \in H\} \\
&= &&H\,g_i^{-1} x_j \,,
\end{aligned}
$$

where in the second to last step we note that the second item in the tuple, $G\,x_0$) is constant (it represents the entire homogeneous space $X = G\,m_0$), and thus we can represent the tuple simply with the first element $H\,g_i^{-1} x_j$. Thus any equivalence class $[x_i, x_j]$ can be represented by the orbit $Hg_i^{-1}x_j$, with $g_i$ any such that $x_i = g_i\,x_0$. $\qquad\square$

### B.2 PROOF OF THEOREM 1

Theorem 1 lists four cases of equivalence classes and provides a bijective map for each of them:

$$\mathbb{R}^2 \text{ and } \mathbb{R}^3 : \qquad [\mathbf{p}_i, \mathbf{p}_j] \;\mapsto\; a_{ij} = \|\mathbf{p}_j - \mathbf{p}_i\| \,, \tag{A3}$$

$$\mathbb{R}^2 \times S^1 \text{ and SE(2)} : \quad [(\mathbf{p}_i, \mathbf{o}_i), (\mathbf{p}_j, \mathbf{o}_j)] \;\mapsto\; a_{ij} = (\mathbf{R}_{\mathbf{o}_i}^{-1}(\mathbf{p}_j - \mathbf{p}_i), \arccos \mathbf{o}_i^\mathsf{T}\mathbf{o}_j) \,, \tag{A4}$$

$$\mathbb{R}^3 \times S^2 : \qquad [(\mathbf{p}_i, \mathbf{o}_i), (\mathbf{p}_j, \mathbf{o}_j)] \;\mapsto\; a_{ij} = \begin{pmatrix} \mathbf{o}_i^\mathsf{T}(\mathbf{p}_j - \mathbf{p}_i) \\ \|(\mathbf{p}_j - \mathbf{p}_i) - \mathbf{o}_i^\mathsf{T}(\mathbf{p}_j - \mathbf{p}_i)\mathbf{o}_i\| \\ \arccos \mathbf{o}_i^\mathsf{T}\mathbf{o}_j \end{pmatrix} \,, \tag{A5}$$

$$\text{SE(3)} : \qquad [(\mathbf{p}_i, \mathbf{R}_i), (\mathbf{p}_j, \mathbf{R}_j)] \;\mapsto\; a_{ij} = (\mathbf{R}_i^{-1}(\mathbf{p}_j - \mathbf{p}_i), \mathbf{R}_i^{-1}\mathbf{R}_j) \,. \tag{A6}$$

We provide a constructive proof for each of these cases by providing an inverse mapping $a_{ij} \mapsto [\mathbf{p}_i, \mathbf{p}_j]$. This boils down to defining representative $x_{ij}$ of the orbit of $H\, g_i^{-1} x_j$ for a given $a_{ij}$. The proofs are as follows.

*Proof.* For the $\mathbb{R}^n$ case the orbit representative can be given given by

$$x_{ij} = a_{ij}\, \mathbf{e}_z \,, \tag{A7}$$

with $\mathbf{e}_z = \left(\begin{smallmatrix} 0 \\ 0 \\ 1 \end{smallmatrix}\right)$. The equivalence class is represented by the orbit $H\, g_i^{-1} x_j = H\, \mathbf{R}_i^{-1}(\mathbf{p}_j - \mathbf{p}_i) = H\, \mathbf{p}_{ij}$, with $\mathbf{p}_{ij} = \mathbf{p}_j - \mathbf{p}_i$. Thus $\mathbf{p}_{ij}$ is a valid representative. Since $H$ does not alter the norm of $\mathbf{p}_{ij}$ and it acts transitively on spheres of a given radius, we can say that $x_{ij}$ lies in the orbit since $\|x_{ij}\| = \|a_{ij}\mathbf{e}_z\| = \|\mathbf{p}_{ij}\|$. This proofs that the mapping in (A3) is bijective, with (A7) as inverse mapping.

To prove the $\mathbb{R}^2 \times S^1$, $SE(2)$, and $SE(3)$ cases we first remark that $\mathbb{R}^2 \times S^1$ is equivalent to the $SE(2)$. We can uniquely identify any $(\mathbf{p}, \mathbf{o}) \in \mathbb{R}^2 \times S^1$ with a roto-translation $(\mathbf{p}, \mathbf{R}_\mathbf{o}) \in \mathrm{SO}(2)$ via $(\mathbf{p}, \mathbf{o}) = (\mathbf{p}, \mathbf{R}_\mathbf{o})(\mathbf{0}, \mathbf{e}_x)$ with $\mathbf{e}_x = \left(\begin{smallmatrix} 1 \\ 0 \end{smallmatrix}\right)$, noting that the stabilizer group $\mathrm{Stab}_{\mathrm{SO}(2)}(\mathbf{0}, \mathbf{e}_x) = \{(\mathbf{0}, \mathbf{I})\}$ is trivial. Thus the spaces are equivalent. Since in these three cases the stabilizer group $H = \{e\}$ is trivial, there is only one unique group element associated with each point $x_i \equiv g_i$ and $x_j \equiv g_j$, with $g_i, g_j$ such that $x_i = g_i x_0$ and $x_j = g_j x_0$. And thus the orbit that represents the equivalence class $[x_i, x_j] \equiv H\, g_i^{-1} x_i = \{g_i^{-1} g_j\}$ consist of a single element. We can write its representative

$$x_{ij} = g_i^{-1} g_j \,, \tag{A8}$$

which is precisely the relative group action for SE(2) in (A4) and for SE(3) in (A6). This proofs that (A4) and (A6) are bijective with (A8) as inverse mapping.

The representative for the $\mathbb{R}^3 \times S^2$ case is given by

$$x_{ij} = \left(\begin{pmatrix} b \\ 0 \\ a \end{pmatrix}, \begin{pmatrix} \sin c \\ 0 \\ \cos c \end{pmatrix}\right) \,, \tag{A9}$$

with $a, b, c$ the components of the attribute where $a_{ij} = (a, b, c)^T$.

Note that the orbits are given by $H\, (\mathbf{R}_{\mathbf{o}_i}^{-1}\mathbf{p}_{ij}, \mathbf{R}_{\mathbf{o}_i}^{-1}\mathbf{o}_j)$. Further note that rotations in $H$ around the $\mathbf{e}_z$ axis act on three independent subspaces of $\mathbb{R}^3 \times S^1$. The spatial part of which is given by $V_a = \mathrm{span}\{\mathbf{e}_z\}$, and $V_b = \mathrm{span}\{\mathbf{e}_x, \mathbf{e}_y\}$, with $\mathbf{e}_x = \left(\begin{smallmatrix} 1 \\ 0 \\ 0 \end{smallmatrix}\right)$ and $\mathbf{e}_y = \left(\begin{smallmatrix} 0 \\ 1 \\ 0 \end{smallmatrix}\right)$. We project the chosen representative onto this basis, such that $\mathbf{R}_{\mathbf{o}_i}^{-1}\mathbf{p}_{ij} = x\,\mathbf{e}_x + y\,\mathbf{e}_y + z\,\mathbf{e}z$. First compute $z$ as

$$z = (\mathbf{R}_{\mathbf{o}_i}^{-1}\mathbf{p}_{ij})^T \mathbf{e}_z = \mathbf{p}_{ij}^T \mathbf{R}_{\mathbf{o}_i}^T = \mathbf{p}_{ij}^T \mathbf{o}_i = a \,.$$

The vector $a\mathbf{e}_z$ is invariant to the action of $H$. The $z$ component of the unit vector in $S^2$ is also invariant, and thus uniquely determined as

$$(\mathbf{R}_{\mathbf{o}_i}^{-1}\mathbf{n}_j)^T \mathbf{e}_z = \mathbf{n}_j^T \mathbf{n}_i = \cos c,$$

and thus $c = \arccos \mathbf{n}_i^T \mathbf{n}_j$. Since both quantities $a$ and $c$ are obtained independent of a chosen rotation in $H$, we can pick any $b$ as long as the vector $\left(\begin{smallmatrix} b \\ 0 \\ a \end{smallmatrix}\right)$ lies in the orbit $H\, \mathbf{R}_{\mathbf{o}_i}^{-1}\mathbf{p}_{ij}$. Since $H$ is norm preserving, we only have to check that for the chosen $b$ the norm $\|(b, 0)^T\| = b$ coincides with the norm of the $V_b$ vector component of $\mathbf{R}_{\mathbf{o}_i}^{-1}\mathbf{p}_{ij}$. The component of $\mathbf{R}_{\mathbf{o}_i}^{-1}\mathbf{p}_{ij}$ in $V_b$ is given by

$(\mathbf{R}_{\mathbf{o}_i}^{-1}\mathbf{p}_{ij}) - (\mathbf{p}_{ij}^T\mathbf{o}_i)\mathbf{e}_z$, and its norm by

$$\|(\mathbf{R}_{\mathbf{o}_i}^{-1}\mathbf{p}_{ij}) - (\mathbf{p}_{ij}^T\mathbf{o}_i)\mathbf{e}_z\| = \|(\mathbf{R}_{\mathbf{o}_i}\mathbf{R}_{\mathbf{o}_i}^{-1}\mathbf{p}_{ij}) - (\mathbf{p}_{ij}^T\mathbf{o}_i)\mathbf{R}_{\mathbf{o}_i}\mathbf{e}_z\| = \|(\mathbf{p}_{ij}) - (\mathbf{p}_{ij}^T\mathbf{o}_i)\mathbf{o}_i\| = b\,,$$

where in the second step we used that the norm is invariant under any rotation in $SO(3)$, and thus we are free to multiply the vector in the norm with $\mathbf{R}_{\mathbf{o}_i}$. In conclusion, the components $a$ and $c$ were uniquely determined the relative group action $(\mathbf{R}_{\mathbf{o}_i}^{-1}\mathbf{p}_{ij}, \mathbf{R}_{\mathbf{o}_i}^{-1}\mathbf{o}_j)$ and $b$ had to be chosen to correspond to the radius of the orbit in the $xy$ plane. This proves that (A5) is bijective, with (??) as inverse mapping.

For all cases, we have now provided valid representatives, and thus for each provided a mapping from attribute to orbit. And thus the provided equivalence class to attribute mappings $[x_i, x_j] \mapsto a_{ij}$ are bijective. $\qquad\square$

### B.3    Proof of Corollary 1.1

Corollary 1 implies that the PΘNITA architectures are equivariant universal approximators. In order to prove this we first show that each of the steps in the architecture is indeed equivariant, and then make use of the results in Dym & Maron (2020); Villar et al. (2021); Gasteiger et al. (2021) to prove universality.

#### B.3.1    Equivariance of Message Passing Layers with Invariant Attributes

First, we make precise the notion of equivariance in the point cloud setting (see Sec. A for the bundle vs point cloud viewpoint). Let us consider a point cloud in a space $X$ which we denote with $\mathcal{G} = (\mathcal{V}, \mathcal{E}, \mathcal{X}, \mathcal{F})$, in which $\mathcal{X} \subset X = \{x_i \mid i \in \mathcal{V}\}$ denotes the set of points associated with the nodes $i$, and $\mathcal{F} = \{f_i \in V \mid i \in \mathcal{V}\}$ denotes the set node features that live in a vector space $V_\rho$, with $\rho$ the representation of $SO(n)$ that acts on $V_\rho$. Let the action of $g = (\mathbf{x}, \mathbf{R}) \in SE(n)$ on the graph be denoted with

$$g\,\mathcal{G} = (\mathcal{V}, \mathcal{E}, g\,\mathcal{X}, g\,\mathcal{F})\,,$$

with

$$g\,\mathcal{X} = \{g\,x_i \mid i \in \mathcal{V}_\downarrow\}\,,$$
$$g\,\mathcal{F} = \{\rho(\mathbf{R})\,f_i \mid i \in \mathcal{V}\}\,.$$

Then a graph message passing layer $MPN$ is said to be equivariant to the group $SE(n)$ if

$$MPN(g\,\mathcal{G}) = g\,MPN(\mathcal{G})\,.$$

**Proposition 1.1.** *Message passing layers as described by Eqs 2-4 as conditioned on the attributes in Theorem 1 are $SE(n)$ equivariant.*

*Proof.* The layers perform a regular group convolution (cf Sec. A) in which case the feature spaces are considered to be vectors of scalars in $\mathbb{R}^c$ on which rotations act trivially, i.e., $\rho(\mathbf{R})f_i = f_i$. Since the positions of the point clouds are un-altered in the message passing steps the output point cloud $\mathcal{X}'$ equals the input point cloud $\mathbf{X}$, and since the attributes $[g\,x_i, g\,x_j] \mapsto a_{ij}$ are invariant for all $g \in SE(n)$, all steps in the message passing scheme are invariant to the group actions. Let us denote the input and output graphs as $G^{in} = (\mathcal{V}, \mathcal{E}, \mathcal{X}, \mathcal{F}^{in})$ and $G^{out} = (\mathcal{V}, \mathcal{E}, \mathcal{X}, \mathcal{F}^{out})$. Then we specifically have that

$$g\,G^{in} \mapsto g\,G^{out} \quad\Leftrightarrow\quad (\mathcal{V}, \mathcal{E}, g\,\mathcal{X}, \mathcal{F}^{in}) \mapsto (\mathcal{V}, \mathcal{E}, g\,\mathcal{X}, \mathcal{F}^{out})\,.$$

So, the node features are invariant and the graph itself (point cloud) is not updated by the message passing layer and thus is trivially equivariant. $\qquad\square$

#### B.3.2    Equivariance of Lifting The Graph from $\mathbb{R}^n$ to $\mathbb{R}^n \times S^{n-1}$

Now consider a lifting transform that assigns an (approximately) uniform grid of elements in $S^{n-1}$ to each based point in a point cloud in $X = \mathbb{R}^n$. Let such a grid be denoted with $\mathcal{S} \subset S^{n-1}$. For now let us consider the availability of an infinite dimensional grid $\mathcal{S} = S^{n-1}$, as in the fiber bundle viewpoint of Sec. A, and that we are thus able to assign to every point $\mathbf{p} \in \mathcal{X}$ and every $\mathbf{o} \in S^{n-1}$ a feature vector through the point-wise spherical signals $f_{\mathbf{x}} : S^{n-1}$.

We then consider lifting of a graph $G = (\mathcal{V}, \mathcal{E}, \mathcal{X}, \mathcal{F})$ with $\mathcal{X} \subset \mathbb{R}^n$ to a graph $G^{\uparrow} = (\mathcal{V}, \mathcal{E}, \mathcal{X}^{\uparrow}, \mathcal{F}^{\uparrow})$ via the `ScalarToSphere` and `VecToSphere` modules. The lifted graph $G^{\uparrow}$ can either be seen as a point cloud in $X^{\uparrow} \subset \mathbb{R}^n \times S^{n-1}$ with scalar features $f_i \in \mathbb{R}^c$, or as a (steerable) feature field over the positions in $\mathcal{X} \subset \mathbb{R}^n$ and with spherical signals $f_i \in V_{\rho^L_{\mathbb{L}_2(S^{n-1})}}$ as node features. We denote the latter with $G^{\rho} = (\mathcal{V}, \mathcal{E}, \mathcal{X}, \mathcal{F}^{\rho})$.

We adopt the bundle viewpoint and consider lifting to the graph $G^{\rho}$. Scalar features $s_i$ at node $i \in \mathcal{V}$ will be lifted via

$$f_i(\mathbf{o}) = \texttt{ScalarToSphere}[s_i](\mathbf{o}) = s_i \,,$$

i.e., it generates constant signals over $S^{n-1}$ with value $s_i$. Vector features $\mathbf{v}_i$ at node $i$ are lifted to a spherical signal via

$$f_i(\mathbf{o}) = \texttt{VecToSphere}[\mathbf{v}_i](\mathbf{o}) = \mathbf{v}_i^T \mathbf{o} \,.$$

**Proposition 1.2.** *Lifting a graph $G$ to a graph $G^{\rho}$ via the `VecToSphere` and `ScalarToSphere` operations, is equivariant via*

$$(\mathcal{V}, \mathcal{E}, \mathcal{X}, \mathcal{F}) \mapsto (\mathcal{V}, \mathcal{E}, \mathcal{X}, \mathcal{F}^{\rho}) \Rightarrow$$
$$(\mathcal{V}, \mathcal{E}, g\,\mathcal{X}, g\,\mathcal{F}) \mapsto (\mathcal{V}, \mathcal{E}, g\,\mathcal{X}, g\,\mathcal{F}^{\rho}) \,.$$

*That is, the nodes in the lifted graph are equally transformed by the action of $g \in SE(n)$ and the spherical signals $f_i \in \mathbb{L}_2(S^{n-1})$ are permuted via the left-regular representation $\rho$ via $g\,\mathcal{F}^{\rho} = \{\rho(g)f_i \mid i \in \mathcal{V}\}$.*

*Proof.* For the `ScalarToSphere` operation we have that the spherical signals $f_i(\mathbf{o}) = s_i$ are invariant since the scalars in the original graph are unaffected by the group action. For the `VecToSphere` operation we have that

$$\mathbf{R}\mathbf{v}_i \mapsto \texttt{VecToSphere}[\mathbf{R}\mathbf{v}_i](\mathbf{o}) = \mathbf{v}_i^T \mathbf{R}^T \mathbf{o} = (\rho^L_{\mathbb{L}_2(S^{n-1})}(\mathbf{R})\,\texttt{VecToSphere}[\mathbf{v}_i])(\mathbf{o}) \,,$$

and thus a rotation of the input vectors leads to a permutation by $\rho^L_{\mathbb{L}_2(S^{n-1})}(\mathbf{R})$ of the spherical signals. $\square$

### B.3.3 EQUIVARIANCE OF PREDICTING SCALARS AND VECTORS

From the lifted graph $G^{\rho}$ we can predict at each position $\mathbf{x}_i \in \mathcal{X}$ a scalar or a vector using the `SphereToScalar` and `SphereToVec` modules. In their continuous forms they are given by

$$s_i = \texttt{SphereToScalar}[f_i] = \int_{S^{n-1}} f_i(\mathbf{o}) \,, \tag{A10}$$

$$\mathbf{v}_i = \texttt{SphereToVec}[f_i] = \int_{S^{n-1}} f_i(\mathbf{o})\,\mathbf{o}\,\mathrm{d}\mathbf{o} \,. \tag{A11}$$

**Proposition 1.3.** *The process of predicting scalars via Eq. (A10) is invariant and predicting vectors via Eq. (A11) is $SE(n)$ equivariant via*

$$\texttt{SphereToVec}[\rho(\mathbf{R})f_i] = \mathbf{R}\,\texttt{SphereToVec}[f_i] \,.$$

*Proof.* Since the scalars are invariant to rotations, the `SphereToScalar` module is trivially invariant. For the `SphereToVec` module we have that if the input spherical signals were to be transformed via $\rho(\mathbf{R})$, than we have

$$
\begin{aligned}
\rho(R)f_i \;\mapsto\; \texttt{SphereToVec}[\rho(\mathbf{R})f_i] &= \int_{S^{n-1}} \rho(\mathbf{R})f_i(\mathbf{o})\mathbf{o}\,\mathrm{d}\mathbf{o} \\
&= \int_{S^{n-1}} f_i(\mathbf{R}^T\mathbf{o})\mathbf{o}\,\mathrm{d}\mathbf{o} \\
&\overset{*}{=} \int_{S^{n-1}} f_i(\mathbf{o})\mathbf{R}\mathbf{o}\,\mathrm{d}\mathbf{o} \\
&= \mathbf{R}\int_{S^{n-1}} f_i(\mathbf{o})\mathbf{o}\,\mathrm{d}\mathbf{o} \\
&= \mathbf{R}\,\texttt{SphereToVec}[f_i] \,,
\end{aligned}
$$

where in the step $\overset{*}{=}$ we used the change of variables $\mathbf{o} \to \mathbf{R}\,\mathbf{o}$. $\square$

### B.4 Proof of Corollary 1.1

Corollary 1.1 states the following: Message passing networks in geometric graphs over $\mathbb{R}^n \times S^{n-1}$ and $SE(n)$, with message functions conditioned on the attributes of Thm. 1, are equivariant universal approximators. The proof is as follows.

*Proof.* In the above, we have shown that all the steps in the PΘNITA architecture are equivariant, and it allows us to predict scalar or vector fields. As discussed in Sec. A, the PΘNITA architecture and its layers fit the steerable tensor-field network class of equivariant graph neural networks. Dym & Maron (2020) have proven such types of equivariant graph neural networks to be equivariant universal approximators. As such, (Dym & Maron, 2020, Theorem 2) proves that the bundle implementation of PΘNITA , as described in the main body of this paper, is indeed an equivariant universal approximator. In the case of the point cloud approach (Sec. A.4), universality is given by (Gasteiger et al., 2021, Theorem 3), which is based on the results in (Villar et al., 2021). □

## C  Implementation

### C.1  Used library

Our implementations are done in Pytorch (Paszke et al., 2019), using Pytorch-Geometric's message passing and graph operations modules (Fey & Lenssen, 2019), and made use of WandB (Biewald, 2020) for logging. Our code is available at `https://github.com/ebekkers/ponita` .

### C.2  Spherical grids

**Uniform grids on $S^2$.**   As described in the main body, we assign a spherical grid to each node in the graph. The grid consists of $N$ points which cover the sphere as uniformly as possible. Note that an exact uniform grid can only be achieved in five cases, corresponding to the five platonic solids with $N = 4, 6, 8, 12, 20$. To achieve an (approximate) uniform grid for any possible $N$, and thus have full flexibility in specifying angular resolution, we generate grids via a repulsion method. This method randomly initializes points on the sphere and then via gradient descent iteratively pushes them away from each other until convergence. We used the repulsion model code of Kuipers & Bekkers (2023), which was used in their work to generate uniform grids on SO(3).

**Random grids.**   To minimize a potential bias to particular directions in the grid, we randomly perturb the grids with a random rotation matrix in each forward pass of the method. Herein we follow the approach of Knigge et al. (2022) for achieving equivariance to continuous groups, sampled on discrete grids. Every graph in the batch obtains its own randomly rotated spherical grid. This is akin to rotational data augmentation.

**Grid resolution**   We found that a grid resolution of $N = 12$ was generally sufficient for all tasks. However, we got slightly better results with $N = 20$, and thus ran all results with this setting. Increasing beyond $N = 20$ did not seem to improve results.

### C.2.1  Attribute embedding

The continuous kernels used in PΘNITA can be considered as Neural Fields. For Neural Fields, it is common to first embed/vectorize the input coordinates. A common approach for this is the Random Fourier Feature embedding, which samples random Fourier basis functions on the provided input coordinates (Tancik et al., 2020). We experimented with this approach as well but found that a simple Polynomial Embedding worked slightly better and was less sensitive to chosen hyperparameters (polynomial degree). The polynomial embedding modules take as input a vector of values $(x, y, z, \dots)$ and generate polynomial combinations of the inputs up to a maximum degree. I.e. `PolynomialEmbed`$(x, y, z) = (x, y, z, xy, xz, yz, xxy, xxz, \dots)$.

In all experiments, the kernel used a shared basis which was computed at the start of the forward pass

$$a_{ij} \rightarrow \boxed{\text{PE}} \rightarrow \boxed{\text{Linear}} \rightarrow \boxed{\text{ActFn}} \rightarrow \boxed{\text{Linear}} \rightarrow \boxed{\text{ActFn}} \rightarrow b_{ij}\,,$$

and each group convolution layer then obtains its layer-specific kernel from this shared basis via

$$b_{ij} \rightarrow \boxed{\texttt{Linear}} \rightarrow k_{ij}^l .$$

In all layers, we used the Gaussian Error Linear Unit (`GeLU`) as an activation function. In all experiments, we used a 256-dimensional basis.

# D  EXPERIMENTAL DETAILS

In this section, we provide the hyperparameters used for training the models and provide additional information regarding the benchmarks and the results.

## D.1  MD17 EXPERIMENTS

**Training settings.**  The rMD17 results were obtained with PΘNITA and PNITA with $L = 5$ layers, $C = 128$ hidden features. The polynomial degree was set to 3. The models were trained for 5000 epochs, with a batch size of 5. We used the Adam optimizerKingma & Ba (2014), with a learning rate of $5e-4$, and with a CosineAnealing learning rate schedule with a warmup period of 50 epochs. We used $N = 20$ grid points on the sphere. The networks optimized the following loss

$$\mathcal{L} = \lambda_E ||\hat{E} - E||^2 + \lambda_F \frac{1}{3M} \sum_{i=1}^{M} \sum_{\alpha=1}^{3} || - \frac{\partial \hat{E}}{\partial \mathbf{p}_{i,\alpha}} - F_{i,\alpha}||^2 \qquad (A12)$$

with $\mathbf{p}_i \in \mathbb{R}^3$. The loss is a weighted sum of energy and force loss terms. Here $M$ is the number of atoms in the system, $\hat{E}$ is the predicted potential energy and $\lambda_E$ and $\lambda_F$ are the energy- and force-weightings, respectively. We set $\lambda_F = 500$. The embedding layer is a linear embedding of the one-hot encoded atom numbers, followed by the `ScalareToSphere` module. As explained in the main body, the readout layer was applied after every ConvNeXt block and consisted of a single Linear layer.

## D.2  DENOISING DIFFUSION MODELS

### D.2.1  SUMMARY OF THE DENOISING DIFFUSION PROBLEM

In diffusion processes, distributions are learned through a reverse diffusion process, i.e. a denoising process. Consider point clouds $\boldsymbol{x} = (\boldsymbol{x}_1, \ldots, \boldsymbol{x}_M) \in \mathbb{R}^{M \times 3}$ with corresponding features $\boldsymbol{h} = (\boldsymbol{h}_1, \ldots, \boldsymbol{h}_M) \in \mathbb{R}^{M \times \text{nf}}$, the diffusion process that adds noise $z_t$ for time $t = 0, \ldots T$ is given by a multivariate normal distribution:

$$q(\boldsymbol{z}_t \mid \boldsymbol{x}) = \mathcal{N}(\boldsymbol{z}_t \mid \alpha_t \boldsymbol{x}_t, \sigma_t^2 \mathbf{I}),$$

where $\alpha_t \in \mathbb{R}^+$ controls signal retention and $\sigma_t$ controls noise added. Sohl-Dickstein et al. (2015); Ho et al. (2020) shows a special case of variance-preserving noising process where $\alpha_t = \sqrt{1 - \sigma_t^2}$. Kingma et al. (2021); Hoogeboom et al. (2022) define signal to noise ratio as $SNR(t) = \frac{\alpha_t^2}{\sigma_t^2}$. Taking into account that the diffusion process is Markov, the entire noising process is then written as:

$$q(\boldsymbol{z}_0, \boldsymbol{z}_1, \ldots, \boldsymbol{z}_T \mid \boldsymbol{x}) = q(\boldsymbol{z}_0 \mid \boldsymbol{x}) \prod_{t=1}^{T} q(\boldsymbol{z}_t \mid \boldsymbol{z}_{t-1}).$$

The generative process in a diffusion model is defined with respect to the true denoising process, and the variable $x$ is approximated using a neural network. Following the findings of Ho et al. (2020); Kingma et al. (2021), the variational lower bound on the log-likelihood term of $x$ is given by

$$\mathcal{L}_t = \mathbb{E}_{\boldsymbol{\epsilon} \sim \mathcal{N}(\mathbf{0}, \mathbf{I})} \left[ \frac{1}{2}(1 - \text{SNR}(t-1)/\text{SNR}(t)) \|\boldsymbol{\epsilon} - \hat{\boldsymbol{\epsilon}}\|^2 \right]$$

where we predict a gaussian noise and use the parameterization $\boldsymbol{z}_t = \alpha_t \boldsymbol{x} + \sigma_t \boldsymbol{\epsilon}$, then the neural network $\phi$ outputs $\hat{\boldsymbol{\epsilon}} = \phi(\boldsymbol{z}_t, t)$.

Köhler et al. (2020) showed that invariant distributions composed with an equivariant invertible function result in an invariant distribution. Additionally, Xu et al. (2022) proved that for $x \sim p(x)$ that is invariant to a group $G$, and the transition probabilities of a Markov chain defined as $y \sim p(y \mid x)$ are equivariant, then the marginal distribution of $y$ at any time step is invariant to group transformations. In the case of the denoising diffusion model, we need an invariant distribution and the neural network parameterizing the denoising diffusing process to be equivariant. This results in the marginal distribution of the denoising model to be an invariant distribution. For detailed proof, we refer to Xu et al. (2022).

In order to define the diffusion process, we take $\alpha_t = \sqrt{1 - \sigma_t^2}$ as per Sohl-Dickstein et al. (2015) and let $\alpha_t = (1 - 2s).f(t) + s$ where $f(t) = (1 - (t/T)^2)$, such that values decrease monotonically, starting $\alpha_0 \approx 1$ and $\alpha_T \approx 0$, similar to EDMs for a fair comparison. To avoid numerical instabilities, during sampling, we follow a clipping procedure similar to Dhariwal & Nichol (2021) and compute $\alpha_{t|t-1} = \alpha_t / \alpha_{t-1}$, where $\alpha_{-1} = 1$ and values of $\alpha_{t|t-1}^2$ are clipped from below by .001 similar to Hoogeboom et al. (2022).

### D.2.2 EXPERIMENTAL DETAILS

**Training settings.** All hyperparameters were set the same as in the original EDM implementation Hoogeboom et al. (2022). That is, we used $L = 9$ layers, $C = 256$ hidden features. The models were trained for 1000 epochs (deviationg from the 3000 as the EDM baseline does), with batch size of 64. The polynomial degree of the basis was set to 3. We used the Adam optimizerKingma & Ba (2014), with a learning rate of $5e - 4$, and without any learning rate scheduler. We used $N = 20$ grid points on the sphere.

The embedding layer is a linear embedding of the one-hot encoded atom numbers, followed by the `ScalareToSphere` module. The readout was a single linear layer, applied to the output of the last ConvNext block, and produced a single channel velocity signal on the sphere. This signal was converted to a vector via the `SphereToVec` module.

### D.3 N-BODY EXPERIMENT

**Training settings.** We trained PΘNITA with $L = 5$ layers, $C = 128$ hidden features. The polynomial degree was set to 3. The models were trained for $10,000$ epochs, with batch size of 100. We used the Adam optimizerKingma & Ba (2014), with a learning rate of $5e - 4$, and with a CosineAnealing learning rate schedule with a warmup period of 100 epochs. We used $N = 20$ grid points on the sphere.

The embedding layer took two vector features as inputs initial velocity and a direction vector pointing from the particle position to the average position of all particles. These two vectors were embedded as two scalar fields over the sphere via the `VecToSphere` module. The embedding also took two scalar features as input: charge (-1 or +1) and the norm of the initial velocity. Both were embedded on the sphere via the `ScalarToSphere` module. The 4 features were mapped to a vector of size $C$ (the hidden dimension size). The readout layer was applied after every ConvNext block, and conisted of a Linear layer followed by the `SphereToVec` module. The final predicted velocity was the average of the velocity predictions after each layer.

In addition to the invariant geometric attributes, the basis functions also took the product the charges of the sending and receiving node as input.

## E ADDITIONAL EXPERIMENTS

### E.1 SUPER-PIXEL MNIST

To verify the impact of lifting point clouds to position orientation space in 2D, we benchmark PΘNITA on superpixel mnist. Superpixel mnist is a dataset of 2D point clouds of MNIST digits, in which each of the original images is segmented into 75 superpixels (clusters of similar pixels). We use the data loader as provided by the library torch-geometric Fey & Lenssen (2019). We utilize the exact same architecture as in all the other experiments presented so far, except that now we utilize the three 2D invariants as given in (8).

We used the following hyper parameters. We trained for 50 epochs with a batch size of 96, a cosine learning rate decay, and initial learning rate of $5e-4$. For PΘNITA we used $N = 10$ grid points to sample the circle.

We compare our results with PNITA and PΘNITA in Table 4 against the recent state-of-the-art method of Finzi et al. (2020b) called Probabilistic Numeric Convolutional Neural Networks (PNCNN), as well as classic graph NN approaches MONET (Monti et al., 2017), SplineCNN (Fey et al., 2018), GCCP (Walker & Glocker, 2019), and GAT (Avelar et al., 2020). Our results are averaged over 3 runs with different seeds and are reported with the standard deviation in test error over these runs.

Apart from showing that with PΘNITA state-of-the-art results are obtained with the simple group convolutional architecture PΘNITA , the results in Table 4 also show the significant benifit of utilizing group convolutions (PΘNITA ) over planar convolutions with isotropic kernels (PNITA ). The results further show that a simple fully convolutional architecture is sufficient to obtain outstanding results.

Table 4: Classification Error on the 75-SuperPixel MNIST problem.

| Method | Error rate |
|---|---|
| MONET | 8.89 |
| SplineCNN | 4.78 |
| GCGP | 4.2 |
| GAT | 3.81 |
| PNCNN | $1.24_{\pm.12}$ |
| PNITA | $3.04_{\pm.09}$ |
| PΘNITA | $\mathbf{1.17}_{\pm.11}$ |

Table 5: QM9 regression results.

| Target | Unit | DimeNet++ | PNITA | PΘNITA (point cloud) | PΘNITA (fiber bundle) |
|---|---|---|---|---|---|
| $\mu$ | D | 0.0286 | 0.0207 | **0.0115** | 0.0121 |
| $\alpha$ | $a_0^3$ | 0.0469 | 0.0602 | 0.0446 | **0.0375** |
| $\epsilon_{HOMO}$ | meV | 27.8 | 26.1 | 18.6 | **16.0** |
| $\epsilon_{LUMO}$ | meV | 19.7 | 21.9 | 15.3 | **14.5** |
| $\Delta\epsilon$ | meV | 34.8 | 43.4 | 33.5 | **30.4** |
| $\langle R^2 \rangle$ | $a_0^2$ | 0.331 | **0.149** | 0.227 | 0.235 |
| ZPVE | meV | **1.29** | 1.53 | **1.29** | **1.29** |
| $U_0$ | meV | **8.02** | 10.71 | 9.20 | 8.31 |
| $U$ | meV | **7.89** | 10.63 | 9.00 | 8.67 |
| $H$ | meV | 8.11 | 11.00 | 8.54 | **8.04** |
| $G$ | meV | 0.0249 | 0.0112 | 0.0095 | **0.00863** |
| $c_v$ | $\frac{cal}{mol\,K}$ | 0.0249 | 0.0307 | 0.0250 | **0.0242** |

### E.2 QM9 REGRESSION

To further demonstrate the potency of PΘNITA on 3D point cloud tasks we validate it on the QM9 benchmark (Ramakrishnan R., 2014) on the task of predicting molecular properties from input atomic point clouds and their covalent bonds.

Since QM9 provides a connectivity derived from the covalent bonds, we have an option to treat the molecules as point clouds in position-orientation space. Namely, we can treat every edge, which connects neighboring point $\mathbf{x}_j$ to central node $\mathbf{x}_i$, as a point in position-orientation space $\mathbb{R}^3 \times S^2$ by treating taking the starting point of the edge as position and the normalized direction vector as orientation. That is, for every edge $e = (j, i) \in \mathcal{E}$ we have identify a point $x_e = (\mathbf{p}_i, \frac{\mathbf{p}_j - \mathbf{p}_i}{\|\mathbf{p}_j - \mathbf{p}_i\|}) \in \mathbb{R}^3 \times S^2$. The covalent bonds thus naturally generate a point-cloud in position-orientation space.

Since this point cloud no-longer has a fiber bundle structure (see App. A) we can no longer separate the convolution over positions and orientations separately and so we do the "domain-mixing" in one single convolution, followed by a channel mixing linear layer. So now the convolution is separated over 2 steps, instead of 3 as for the other experiments, as described in Section 3.2. Otherwise, the PΘNITA architecture is held exactly the same. In this experiment we compare convolutions using either the atomic point cloud as input (PNITA) versus using the covalent-bond cloud (edges) as input (PΘNITA). The latter is close to the approach of DimeNet and GemNet (Gasteiger et al., 2019; 2021) which also perform message passing operations between edges. We therefore compare our approach against DimeNet++ (Gasteiger et al., 2020).

We did a hyperparmeter search for $C$ and $L$ for PNITA and PΘNITA separately and found that for PNITA best results were obtained with $C = 128$ and $L = 5$ and for PΘNITA with $C = 256$ and $L = 9$. We further trained for 1000 epochs with a batch size of 96, with a cosine learning rate decay scheduler and initial learning rate of $5e - 4$. For all atoms we constructed a fully connected graph. For reference, we also trained a fiber bundle PΘNITA with $C = 256$, $L = 9$, and $N = 16$.

The results are presented in Table 5.

