# OpenReview forum: "Fast, Expressive $\mathrm{SE}(n)$ Equivariant Networks through Weight-Sharing in Position-Orientation Space"
_ICLR.cc/2024/Conference — ICLR 2024 poster_

### Official Review · Reviewer_7BQC · 2023-10-27

**Soundness:** 3 good
**Presentation:** 2 fair
**Contribution:** 2 fair
**Rating:** 5
**Confidence:** 2

**Summary:**

This work derives point-pair attributes uniquely identifying the equivalence class of the point-pair and uses these attributes for equivariant group convolutional network. Its application on 3D point clouds achieves state-of-the-art performance and scalability.

**Strengths:**

1. Conceptually simple, powerful, and efficient model verified with extensive experiments.
2. A detailed introduction of previous work.

**Weaknesses:**

1. There are so many contents from previous works that I can hardly tell the novelty. From my point of view, only Theorem 1 is new, but it is still quite easy to prove.

2. Though this work is titled with $SE(n)$, it only discusses $n=3$ case.

**Questions:**

1. Does this work have provable expressivity? For example, can it universally approximate continuous equivariant function?

---

> ### Author Response · Authors · 2023-11-16
>
> Thank you for your time and constructive review! We appreciate your recognition for the simplicity and effectiveness of our method, as well as it's thorough discussion of related work. We next respond to your listed weaknesses and questions.
>
> **Comment 1 (contribution)**
>
> The first mentioned weakness is that it is hard to recognize the novelty, given the many previous works. This is a valuable comment and agree that the issues arises from our drive to unify related works through our approach. We do indeed build upon the so-called shoulders of giants and as such thoroughly discuss related work. Our new contribution can however be described on several levels:
> 1. **Conceptual**: the formal notion of weight sharing in the equivariance setting provides a simple recipe that so far did not exist in literature.
> 2. **Practical**: we provide a simple yet effective architecture. Many of previous works either adopt the steerable approach, or an intricate wiring of message passing schemes based on invariants. Our method greatly simplifies the latter approach. Our architecture in itself follows the standard convnet paradigm is predominatly used in computer vision, and make it effectively applicable to point-cloud data, whilst being fully equivariant.
> 3. **Theoretical**: even though theorem 1 might be simple to proof, we think it is an important contribution because by formally deriving identifiers for equivalence classes we have proven that the obtained attributes are in fact all you need to build fully expressive NNs. Based on your other comment, *we now also include a proof of our approach being a universal equivariant approximator*.
>
> The contributions in the intro is therefore slightly updated. Thanks again for your remark, we think it did improve the presentation.
>
> **Comment 2 (not using case $n=2$)**
>
> Though the work is titled with $SE(n)$, it only discusses the $n=3$ case. It is true that we focussed almost exculisivey on $n=3$. We decided to use $n$ instead of only $n=3$ because this is where a unifying view was still missing. With that we mean, $SE(2)$ as a space is equivalent to the position space $\mathbb{R}^2 \times S^1$. Since literature already extensively covers group convolutional methods for $SE(2)$ equivariance, there is not yet much to add to this. Our method still adds conceptually to the field via the viewpoint of weightsharing, and non-linear convolutions. Notably, our method when applied to 2D data would boil down to regular $SE(2)$ convolutions, which in Weiler and Cesa 2019 have been proven to be more effective than steerable convolutions. This motives the use of working with invariants over the use of tensor field neworks (based on irreducible representations). *We deepen the discussion on the impact of regular group convolutions with a newly added appendix A "The Fiber Bundle vs Point Cloud Viewpoint"*.
>
> **Comment 3 (Expresivity)**
> Finally, you had a question on expressivity. As also responded to one of the other reviewers, *we now formally state that our networks are equivariant universal approximators*, that can represent any $SE(n)$ equivariant function. Since we consider equivariance to the continuous group $SE(n)$ this implies continuous functions as well.
>
> *We deeply thank you for this suggestion as we think this addition might further improve the impact of our work.*

---

### Official Review · Reviewer_MJgQ · 2023-10-31

**Soundness:** 3 good
**Presentation:** 2 fair
**Contribution:** 2 fair
**Rating:** 6
**Confidence:** 4

**Summary:**

This paper studies the message passing scheme for signals define on space $\mathbb{R}^3\times S^2$ with SE(3) symmetry, which then can be used to construct neural architectures for pointcloud data, by lifting the pointcloud from $\mathbb{R}^3$ to $\mathbb{R}^3\times S^2$.   In particular, they give the explicit form of an invariant and bijective edge embedding $a_{i,j}$ for two points $x_i,x_j\in \mathbb{R}^3\times S^2$. This edge embedding serves to weight the pairwise message passing.

They argue that the proposed method is more efficient than expressive $SE(3)$ group convolution (which requires integral on $SE(3)$) and meanwhile more expressive than convolution on $\mathbb{R}^3$ (e.g., SchNet). Experiments on potential prediction, molecule generation as well as trajectory prediction show the better performance and efficiency of the method.

**Strengths:**

- the analysis on invariant feature $a_{i,j}$ seems technically sound and solid to me
- the experimental part is extensive

**Weaknesses:**

- As there are many approaches for lifting a pointcloud, it may not be clear why the space chosen may find the best trade-off between efficiency and expressiveness.
- So I think it could make the work stand out if we can find some applications that require to directly deal with signals on $\mathbb{R}^3\times S^2$.
- Also just a minor personal idea: I think the word "weight sharing" makes me think of constructing linear transformation invariant/equivariant to certain symmetry. While here we are actually seeking for the full invariants describing the equivalence class.

**Questions:**

- There are two P$\Theta$NITA columns in Table 1. Should be a typo?

---

> ### Author Response · Authors · 2023-11-16
>
> Thank you for your time and constructive review. We are happy to read your acknowledgement of our technically sound approach and thorough validation. We will respond to your bulleted remarks one by one:
>
> **Bullet 1 (trade-off between efficiency and expresiveness)**
>
> The first bullet is about the choice for position-orientation space as the optimal trade-off between efficiency and expressiveness. To answer this question we first recall to Section 2.3 "Motivation 2: Efficiency and Expresivity – The Homogeneous Space $\mathbb{R}^3 \times S^2$" of our submission. The essence of it is that the most efficient space to operate on is $\mathbb{R}^3$, however, when working with invariants --and thus the standard deep learning paradigm-- one cannot handle directional information. In this case convolution kernels or message passing functions are insensitive to directions as they can only be conditioned on distances.
>
> Then the next smallest space, among the allowed options $\mathbb{R}^3 \times S^2$ and $SE(3) \equiv \mathbb{R}^3 \times SO(3)$ is, is the position-orientation space $\mathbb{R}^3 \times S^2$. Still using the standard deep learning paradigm (with standard activation functions and other layers), one can now in fact build representations that carry directional information. Since now the base space is larger, one can expect a drop in efficiency, however, we by-pass this by using separable convolutions. See table 1 which shows that the $\mathbb{R}^3$ method PNITA takes 5.8 sec per epoch, and the $\mathbb{R}^3 \times S^2$ method PONITA takes 6.5 sec per epoch. This difference is almost insignificant, in particular compared to the full $SE(3)$ steerable tensor-field type methods like NequIP (20.7 sec per epoch).
>
> Finally, we argue one does not necesarily gain more expressive power by using the higher-dimensional space of $SE(3)$ in terms of performance, *based on the newly added proof of universal approximation power* (which we added in response to a comment of reviewer 7BQC). In the new appendix A we discuss the steerable/fiber bundle-theoretic viewpoint of our method. Through conversion to irreps of spherical signals, our multi-channel spherical signal fields can represent $SO(3)$ functions at each node as well.
>
> In summary, our of the three options $\mathbb{R}^3$, $\mathbb{R}^3 \times S^2$, $\mathbb{R}^3 \times SO(3)$, the position-orientation space is the smallest (and thus most efficient) space that still allows for fully expressive equivariant NN design. For the $n=2$ case there are only two options $\mathbb{R}^2$ and $\mathbb{R}^2 \times S^1$, in which the later is most expressive as $\mathbb{R}^2$ does not permit directional information.
>
> **Bullet 2 (applications of $\mathbb{R}^3 \times S^2$)**
>
> The second bullet indicates that the work could stand out with more applications that directly require working with signals on $\mathbb{R}^3 \times S^2$. As concrete applications we now explicitly mention its applicability to diffusion weighted MRI. In this modality MRI scanners directly obtain measurements related to the diffusivity of water molecules in the brain, as measured along different directions at each voxel location. As diffusivity of water is constraint by the presence and direction of neurons in the brain, this type of data is often used to perform neuron tractography. Other applications are curvature penalized geodesic tracking of bloodvessels (or neurons) in medical data. Both the DWI and tracking applications are discussed in the orientation-score section of the related work section. *Here we now make explicit mention of the diffusion weighted MRI application.*
>
> Another application could be the processing of meshes as point clouds. A mesh surface element could be represent by its centroid and it's normal direction, and thus it could be treated as a point cloud in position-orientaiton space. Another application is that of processing of molecules by doing message passing over the edges (each edge has a starting position and a direction towards its neightbor). Dimenet/Gemnet are examples of this. *In Appendix A, we further discuss the connection to the equivariant learning field and now explicitly mention the mesh and molecular edge cloud application.*

---

> > ### Author Response · Authors · 2023-11-16
> >
> > **Bullet 3 (weight-sharing terminology)**
> >
> > The third comment is on the terminology of "weight sharing", which often is discussed in terms of linear transformations that are equivariant or invariant to a certain symmetry. Thank you for this insightful remark! We agree that the term might evoke different associations with different audiences and hence decided to formally define it in Definition 3.3.
> >
> > The following may be familiar to you, but for the sake of completeness we would like to further clarify what we think the connection you have in might may refer to.
> >
> > The notion of symmetries in linear maps in fact motivated our work and is the underlying principle behind the "convolution is all you need" theorems cited in the paper. In context of signal analysis (as an example here we take feature maps over $\mathbb{R}^n$), a linear layer is an integral transform
> > $f^{out}(y) = \int_{\mathbb{R}^n} k(y,x) f(x) {\rm d}x$, parametrized by a two-argument kernel $k(y,x)$. Note that this is like the continuous equivalent of a matrix-vector multiplication $\mathbf{y} = \mathbf{K} \mathbf{x}$ between two finite dimensional vectors, here we consider infitinite dimensional vectors (functions). Now, when constraining this linear map to be translation equivariant, the transform is parametrized by a one-argument kernel via
> > $$f^{out}(y) = \int_{\mathbb{R}^n} k(x-y)f(x) {\rm d}x .$$
> > When the signal is one-dimensional and discretized on a grid, the corresponding matrix $\mathbf{K}$ is a circulant matrix (convolution can be implemented as a matrix-vector multiplication!).
> >
> > What we do in section 2.4 "Conditional Message Passing as Generalized Convolution" we build on this view point and say that any NN based on pair-wise interactions that are conditioned on invariants (like $x-y$) could be considered *non-linear* generalizations of convolutions.
> >
> > Final remark, our construction is different to methods like for example Finzi 2021, which are about finding linear maps that are equivariant to a large class of groups. I.e., for vector spaces (as in MLPs) they derive the matrices that satisfy the equivariance constraint.
> >
> > Finzi, M., Welling, M. and Wilson, A.G.. (2021). A Practical Method for Constructing Equivariant Multilayer Perceptrons for Arbitrary Matrix Groups. *Proceedings of the 38th International Conference on Machine Learning*, in *Proceedings of Machine Learning Research* 139:3318-3328
> >
> > **Typo**
> >
> > Finally, table 1 indeed has a type, the first PONITA was supposed to be PNITA, the position space version of PONITA.

---

### Official Review · Reviewer_3W6S · 2023-11-03

**Soundness:** 3 good
**Presentation:** 3 good
**Contribution:** 3 good
**Rating:** 8
**Confidence:** 3

**Summary:**

When considering a homogeneous space $X\simeq G/H$ and features that are functions $f:X\to \mathbb{R}^C$, equivariant linear layers are convolutions (Equation 1). The authors propose to extend such layers to the case where $X$ is replaced by a discrete geometry (a graph of points of $X$), which allows for equivariant graph convolutions. They remark that the direct generalization (Equation 5) is not intrinsic as it depends on a represent of $x\simeq [g_x]$; they characterize filters that are intrinsic and give applications to quotients of SE(3) and SE(2).

**Strengths:**

The paper is very well written, the problem well posed and meaningful, and the solutions natural. I find the paper very interesting.

**Weaknesses:**

I don't see any

**Questions:**

None

---

> ### Author Response · Authors · 2023-11-16
>
> Thank you for your time and your encouraging review. It was our intention to present a practical and simplified/unifying view on equivariant neural networks and we greatly appreciate your acknowledgement of our efforts.
>
> If there is anything that you would still like to see clarified please let us know.
>
> For your reference, we updated the paper with a formal statement of universal approximation power. We proof this in a new Appendix sections B3 and B4. To assist the proof and deepen the discussion on the relation of our method to literature we present our work from both a bundle theoretic and point cloud viewpoint in the new appendix A.

---

### Meta-Review · Area_Chair_85Hx · 2023-12-18

**Metareview:**

The paper proposes networks for processing pointcloud data by lifting R^3 to space including orientation: the idea is to be more expressive than R^3 to include directional information while being significantly more efficient than lifting to the more expressive SE(3) group. The reviewers largely held positive support of the paper and particularly highlighted the technical and experimental strengths.

**Justification For Why Not Higher Score:**

The paper has a somewhat narrow focus and applicability for 3D point clouds compared to general ICLR audience (although the title claims SE(n), the analysis and experiments are restricted to n=3).

**Justification For Why Not Lower Score:**

The reviewers were largely in agreement about interesting ideas in the paper.

---

### Decision · Program_Chairs · 2024-01-16

Accept (poster)